# The burden and dynamics of hospital-acquired SARS-CoV-2 in England

Ben S. Cooper[1,2 ✉], Stephanie Evans[3], Yalda Jafari[4], Thi Mui Pham[5], Yin Mo[1,2,6,7], Cherry Lim[1,2], Mark G. Pritchard[1,8], Diane Pople[3], Victoria Hall[3], James Stimson[3], David W. Eyre[9,10,11,12], Jonathan M. Read[13], Christl A. Donnelly[8,14,15], Peter Horby[8], Conall Watson[8], Sebastian Funk[4], Julie V. Robotham[3,12,17] & Gwenan M. Knight[4,16,17]

Hospital-based transmission had a dominant role in Middle East respiratory syndrome coronavirus (MERS-CoV) and severe acute respiratory syndrome coronavirus (SARS-CoV) epidemics[1,2], but large-scale studies of its role in the SARS-CoV-2 pandemic are lacking. Such transmission risks spreading the virus to the most vulnerable individuals and can have wider-scale impacts through hospital–community interactions. Using data from acute hospitals in England, we quantify within-hospital transmission, evaluate likely pathways of spread and factors associated with heightened transmission risk, and explore the wider dynamical consequences. We estimate that between June 2020 and March 2021 between 95,000 and 167,000 inpatients acquired SARS-CoV-2 in hospitals (1% to 2% of all hospital admissions in this period). Analysis of time series data provided evidence that patients who themselves acquired SARS-CoV-2 infection in hospital were the main sources of transmission to other patients. Increased transmission to inpatients was associated with hospitals having fewer single rooms and lower heated volume per bed. Moreover, we show that reducing hospital transmission could substantially enhance the efficiency of punctuated lockdown measures in suppressing community transmission. These findings reveal the previously unrecognized scale of hospital transmission, have direct implications for targeting of hospital control measures and highlight the need to design hospitals better equipped to limit the transmission of future high-consequence pathogens.

Hospital transmission had a central role in the spread of Middle East respiratory syndrome coronavirus (MERS-CoV) and severe acute respiratory syndrome coronavirus (SARS-CoV) in human populations[1,2], and multiple reports have indicated that SARS-CoV-2 is capable of spreading efficiently in healthcare settings[3–11] and is associated with poor outcomes[12,13]. However, attempts to fully document the extent of hospital transmission using systematically collected national data or to take a data-driven approach to quantifying the drivers of such transmission are lacking. Addressing these knowledge gaps is important: hospital transmission directly affects patients likely to have multiple factors associated with poor outcomes; it puts healthcare workers (HCWs) at risk and compromises their ability to provide safe patient care; it disrupts service delivery; and it can have a major role in disseminating infection to vulnerable groups in the community. Moreover, because non-pharmaceutical interventions in the community do not affect rates of transmission from infected patients and HCWs in hospitals, hospital transmission can have notable effects on epidemic dynamics during lockdown periods. Understanding such transmission has implications for both continuing epidemics and threats from new variants even in highly vaccinated populations. We use data from 145 English National Health Service (NHS) acute hospital trusts (organizational units containing one or more acute care hospitals), excluding only those caring exclusively for children. These trusts contained 356 hospitals, had a combined bed capacity of about 100,000 (over 98% of the total NHS general and acute care bed capacity in England in 2020) and employed 859,000 full-time equivalent HCWs, 2.5% of the working-age population of England. From 20 March 2020, all such trusts completed a daily situation report that included essential information on the prevalence and incidence of SARS-CoV-2 infection, the number of patients admitted with SARS-CoV-2 infection and staff absences due to SARS-CoV-2. From 5 June 2020, a classification of the likely source of SARS-CoV-2 infection on the basis of European Centre for Disease Prevention and Control

[1]NDM Centre for Global Health Research, Nuffield Department of Medicine, University of Oxford, Oxford, UK. [2]Mahidol-Oxford Tropical Medicine Research Unit, Faculty of Tropical Medicine, Mahidol University, Bangkok, Thailand. [3]HCAI, Fungal, AMR, AMU and Sepsis Division, UK Health Security Agency, London, UK. [4]Centre for Mathematical Modelling of Infectious Diseases, IDE, EPH, London School of Hygiene & Tropical Medicine, London, UK. [5]Julius Center for Health Sciences and Primary Care, University Medical Center Utrecht, Utrecht University, Utrecht, The Netherlands. [6]Division of Infectious Disease, Department of Medicine, National University Hospital, Singapore, Singapore. [7]Department of Medicine, National University of Singapore, Singapore, Singapore. [8]Pandemic Sciences Institute, Nuffield Department of Medicine, University of Oxford, Oxford, UK. [9]Big Data Institute, Nuffield Department of Population Health, University of Oxford, Oxford, UK. [10]Oxford University Hospitals NHS Foundation Trust, Oxford, UK. [11]NIHR Oxford Biomedical Research Centre, University of Oxford, Oxford, UK. [12]NIHR Health Protection Research Unit in Healthcare Associated Infections and Antimicrobial Resistance at University of Oxford in partnership with UKHSA, Oxford, UK. [13]Lancaster Medical School, Lancaster University, Lancaster, UK. [14]Department of Statistics, University of Oxford, Oxford, UK. [15]MRC Centre for Global Infectious Disease Analysis, Department of Infectious Disease Epidemiology, Imperial College London, London, UK. [16]AMR Centre, IDE, EPH, London School of Hygiene & Tropical Medicine, London, UK. [17]These authors contributed equally: Julie V. Robotham, Gwenan M. Knight. ✉e-mail: ben.cooper@ndm.ox.ac.uk

(ECDC) criteria was also required[14]. This was determined by the interval between hospital admission and date of onset of PCR-confirmed infection in hospitalized patients: community-onset infections were defined as those with an interval of 2 d or less; an interval of 3–7 d led to a classification of indeterminate healthcare-associated; intervals of 8–14 d were classified as probable healthcare-associated; and intervals of 15 d or more were classified as definite healthcare-associated. As few patients have hospital stays exceeding 7 d and many nosocomially infected patients will be discharged before testing positive, such definitions necessarily capture only a proportion of hospital-acquired infections[15].

We make use of these data, linked with other national datasets, to infer the number of hospital-acquired infections in England between June 2020 and February 2021, the pathways of nosocomial transmission and factors potentially modulating such transmission, including hospital characteristics, vaccination coverage and prevalence of relevant variants. Using a model coupling hospital and community dynamics, we then explore the consequences of such nosocomial transmission for the effectiveness of community lockdown measures in averting infections.

Between 10 June 2020 and 17 February 2021, a total of 16,950 and 19,355 SARS-CoV-2 infections in hospital inpatients met the criteria for definite and probable healthcare-associated infections, respectively, corresponding to a median (interquartile range) of 1.7 (1.1, 2.5) detected infections per 1,000 occupied bed days. To estimate the total number of hospital-acquired infections, we multiply the recorded number of definite healthcare-associated infections by the reciprocal of the proportion of hospital-acquired infections that we expect to meet these 'definite healthcare-associated' criteria. Using the empirical length-of-stay distribution, the estimated incubation period distribution and the profile of PCR test sensitivity as a function of time since infection[16] (Fig. 1a–c), we estimate that a policy of PCR testing of symptomatic patients would detect 26% (90% credible interval (90% CrI) 21%, 30%) of hospital-acquired infections, with 12% (10%, 14%) of all hospital-acquired infections meeting criteria for definite healthcare-associated infection (Fig. 1d–f). Adding asymptomatic PCR testing on days of stay 3 and 6 (as recommended by national screening guidance in England at the time) increases the proportion detected to 33% (26%, 38%) but does not substantively alter the proportion classified as definite healthcare-associated. Augmenting symptomatic PCR tests with testing for all patients at 7 d intervals (a policy adopted by some hospitals in England) increases the proportion of hospital-acquired infections detected to 44% (39%, 47%), and the proportion classified as definite healthcare-associated to 17% (16%, 18%). These low probabilities for detection and classification as definite healthcare-associated are a consequence of the typically short lengths of patient stay and low PCR sensitivities early in the course of infection (Fig. 1b,c).

Combining these estimates with the number of reported definite healthcare-associated infections, we infer the number of hospital-acquired infections under two sets of assumptions. First, we assume patient testing followed national guidance at the time, which specified testing of symptomatic patients (without retesting) and included asymptomatic testing on two occasions in the first week but none after day 7 postadmission. This provides a plausible lower bound for the chance of identifying hospital-acquired infections and thus an upper bound for the estimated numbers of such infections. Second, we assume testing for all patients at 7 d intervals postadmission in addition to symptomatic testing of patients (the maximal testing policy known to be used in practice). This provides a plausible upper bound for the chance of identifying hospital-acquired infections and thus a lower bound on the estimated numbers of such infections. Using definite healthcare-associated infections only, this yields as an upper bound a mean (90% CrI) estimate for the number of hospital-acquired infections of 143,000 (123,000, 167,000) and a lower bound of 99,000 (95,000, 104,000). During this period there were 9.2 million hospital admissions from 5.0 million individual patients, so we estimate that between 1% and 2% of admissions developed a hospital-acquired SARS-CoV-2 infection. Similar estimates are obtained when using more granular length-of-stay data and in other sensitivity analyses, whereas repeating the analysis using probable and definite healthcare-associated infections yields estimates that are 20–30% higher (Supplementary Information section 2.1).

There is considerable variation in cumulative rates of hospital-associated infection between trusts, with the highest rates seen in the North West NHS region, and the lowest in the South West and London regions (Extended Data Fig. 1). There is a strong positive correlation between rates of definite and probable hospital-associated infections ($r = 0.76$), and weak positive correlation between definite hospital-associated infection and HCW infection ($r = 0.31$), but only a very weak correlation between definite hospital-associated infection and community-acquired infection ($r = 0.16$). Three hospital characteristics are weakly correlated with cumulative rates of definite hospital-associated infection: bed occupancy ($r = 0.25$), availability of single-bedded rooms ($r = -0.39$) and heated volume per bed, a measure of the volume of heated areas of trust buildings divided by the number of beds ($r = -0.34$).

To quantify drivers of transmission to patients and HCWs we link these data to national datasets (Fig. 2e–l), capturing information on hospital characteristics potentially affecting transmission, alongside regional variation in HCW vaccination and prevalence of the Alpha variant. As no direct measurements of hospital ventilation are available, we use hospital building heated volume per bed as a proxy. This analysis is restricted to 96 of the 145 trusts for which complete data are available and uses negative binomial auto-regression models for which the dependent variable is either the weekly number of patients with healthcare-associated infections or the imputed weekly number of HCWs with confirmed SARS-CoV-2 infection. Independent variables are selected on the basis of biological plausibility. Mechanistic considerations inform the parameterization of the dispersion terms and the inclusion of additive effects for exposures to community-acquired patient infections, hospital-acquired patient infections and infected HCWs (Fig. 2, top row), combined with multiplicative effects of trust characteristics (Fig. 2, middle row), HCW vaccine coverage and Alpha variant prevalence (Fig. 2, bottom row). Posterior predictive distributions from fitted models are shown in Fig. 3 and Extended Data Figs. 2 and 3.

Among the additive terms, the strongest predictor of new healthcare-associated infections is the number of patients in the same trust with healthcare-associated infections the previous week (Fig. 3); thus, one patient with a newly identified healthcare-associated infection the previous week is associated with a further 1.07 (95% CrI 0.93, 1.19) hospital-acquired infections in patients the following week (setting variables representing hospital characteristics to their mean values, and in the absence of the Alpha variant or vaccine effects). Additive effects associated with patient exposures to infected HCWs and patients admitted with SARS-CoV-2 are smaller, although the larger number of such exposures increases their contribution to patient infections (Fig. 3f).

Considering multiplicative effects associated with trust characteristics, increased availability of single rooms is associated with reduced incidence of healthcare-associated infections in patients with an incidence rate ratio (IRR) for a 1 s.d. increase in single room availability (corresponding to a 15% increase in the percentage of beds as single rooms) of 0.91 (0.87, 0.97), whereas heated volume per bed is associated with a similar reduction (IRR 0.90 (0.84, 0.97)) for a 1 s.d. increase, corresponding to an increase per bed of 207 m$^3$, and older hospital buildings were also associated with reduced hospital transmission, although in this case 95% CrIs include the null value of 1.00 (IRR 0.96 (0.92, 1.00)) (Fig. 3). These effects were not seen for infections in HCWs. HCW vaccination was associated with substantial reduction in transmission to patients linked to exposures to infected HCWs, and large

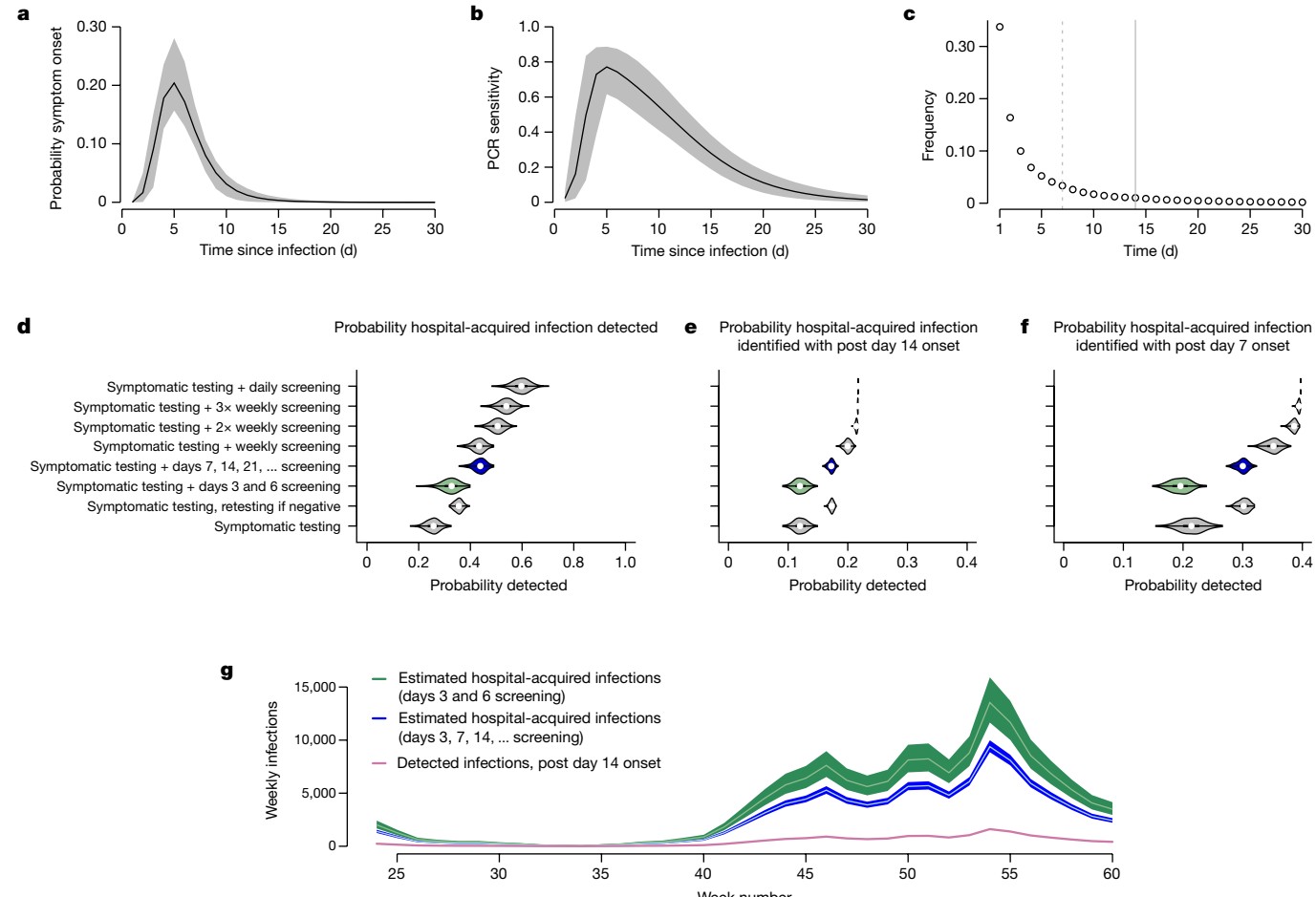

**Fig. 1 | Quantifying the probability of observing hospital-acquired infections and estimating the total number of such infections. a–c**, Model inputs are shown in the top row and include the incubation period distribution[32] (**a**), the PCR sensitivity profile[16] (**b**) and the length-of-stay distribution (**c**) for patients who were not admitted with COVID-19 between June 2020 and February 2021 (solid lines in **a** and **b** show expected values and shaded regions show 95% CrIs). In **c**, the minimum lengths of stays needed to be classified as a probable or definite healthcare-associated infection are shown by dashed and solid vertical lines. **d–f**, Estimates of the probabilities that patients with hospital-acquired SARS-CoV-2 infections have a PCR positive test while in hospital under different screening policies (**d**), and estimates of the probabilities that they both screen positive and meet the post-14 d onset criteria to be considered a 'definite' healthcare-associated infection (**e**) or the post-7 d criteria to be classified as a probable or definite healthcare-associated infection (**f**) are shown in the

middle row. Panels **d–f** are on the basis of 1,000 Monte Carlo samples, with violin plots showing median values (points), interquartile ranges (rectangles) and densities. The Public Health England screening recommendations are highlighted in green and the policy of screening all patients at 7 d intervals after admission is highlighted in blue (note that in contrast to this policy, weekly and 2× and 3× weekly policies screen on fixed days of the week). **g**, The estimated total number of hospital-acquired infections across adult acute NHS trusts in England linked to observed weekly number of detected post-14 d onset infections, assuming the screening policies highlighted in the middle row on the basis of recorded 'definite healthcare-associated infections'; week numbers are counted as 1 plus the number of complete 7 d periods since 1 January 2020. Green and blue shaded regions indicate 90% CrIs and white lines are posterior means.

reductions in the overall rate of infection in HCWs. Increased Alpha variant prevalence was associated with large increases in the rates of infection in both patients and HCWs.

Negative controls can help to assess the likelihood that associations between exposures and outcomes in observational studies result from relationships that are not directly causal (Extended Data Fig. 4)[17]. We use as a negative control outcome the number of patients admitted meeting ECDC definitions for community-acquired SARS-CoV-2 infection. Assuming most hospital admissions with SARS-CoV-2 result from community transmission, this outcome would not be expected to have a strong association with hospital-based exposures. If associations between hospital characteristics (exposures) and this control outcome are similar to those for hospital-acquired infections, it would indicate that confounding is a plausible explanation for observed associations with hospital-acquired infections (for example, owing to differences in hospital characteristics not accounted for in the model). Note, however,

that as some SARS-CoV-2 admissions from the community will result from the readmission of patients infected in hospital, some link is expected. In all models considered with this control outcome, there is no strong association with the number of healthcare-associated infections or with the single room provision, strengthening the evidence that these both play a causal role in the incidence of hospital-acquired infections (Supplementary Tables 15–17). However, both heated volume per bed and HCW vaccination coverage show similar negative associations with the control outcome, as reported for healthcare-associated infection outcomes, indicating the need for caution when considering whether these reported associations might reflect direct causal effects.

To help to interpret estimated regression coefficients we perform a series of simulation studies, generating synthetic transmission datasets from a multitype branching process model, applying an observation model to obtain partially observed infection data and

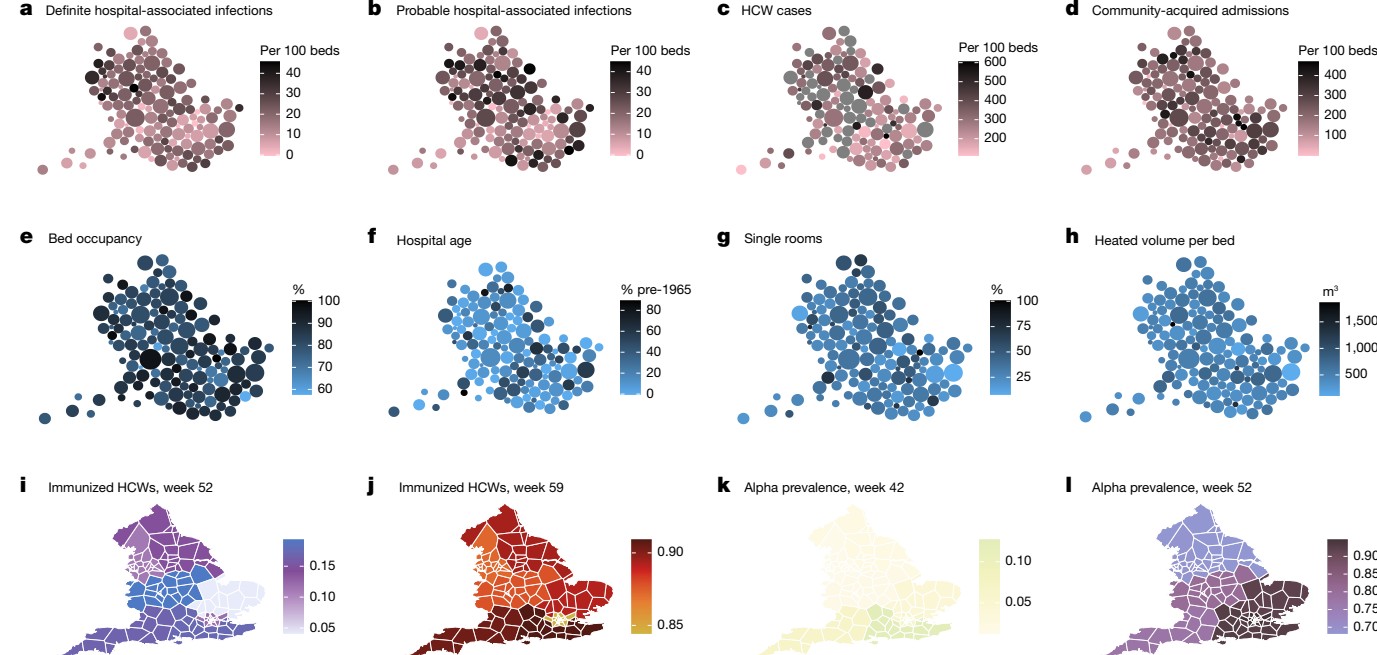

**Fig. 2 | Summary of data used in the analysis. a–d,** Data from situation reports related to SARS-CoV-2 infection in England showing variation between trusts. Each circle corresponds to one NHS trust scaled by the number of available beds. Shading indicates cumulative totals to the end of the period considered (17 February 2021). Geographic locations are approximate. Cumulative number of hospital-associated SARS-CoV-2 infections in patients per 100 hospital beds with first positive sample more than 14 d after admission (**a**); hospital-associated infections in patients with first positive sample more than 7 d after admission (**b**); imputed cumulative number of cases in HCWs (**c**) with grey shading indicating missing data; infections in hospitalized patients with community-onset (**d**). **e–h,** Trust-level data characteristics from the third quarter of 2020: bed

occupancy (**e**); age of acute hospital buildings in the trust expressed as a weighted average of the percentage of hospital buildings constructed in 1964 or earlier, for which weights are the hospital gross internal floor areas (**f**); number of single room beds per trust (including isolation rooms) as a percentage of the number of general and acute beds available in the last quarter of 2020 (**g**); heated volume per bed (**h**). **i–l,** A snapshot of regional HCW immunization data at two time points showing the proportion of HCWs who had received at least one vaccine dose at least 3 weeks earlier at week 52 (**i**) and week 59 (**j**), and regional data on the proportion of PCR-confirmed infections due to the Alpha variant at week 42 (**k**) and week 52 (**l**) (in both cases, Voronoi tessellations centred on the location of the largest hospital in each trust are shown).

replicating the above analysis (Supplementary Information section 2.3). This analysis indicates that when the outcome is patient hospital-acquired infections, regression coefficients typically underestimate the expected number of secondary cases per case when only a proportion of hospital-acquired infections are observed, although represent good approximations as the proportion approaches 1 (Extended Data Fig. 5).

We use estimates from these analyses and the wider literature on hospital-acquired SARS-CoV-2 transmission to inform a dynamic model coupling hospital and community dynamics (Methods and Supplementary Information section 1.2). We consider three scenarios: high hospital transmission, corresponding to self-sustaining within-hospital transmission; and intermediate and low hospital transmission, in which all hospital transmission rates were reduced by 25% and 50%, respectively, compared with the high hospital transmission scenario (Fig. 4). Community transmission rates were identical in all scenarios.

The level of hospital transmission has little overall impact on an unmitigated epidemic or an epidemic controlled by a single lockdown, modelled here as a policy that substantially reduces community transmission (Extended Data Fig. 6). However, when community transmission is controlled through punctuated lockdowns, the extent of hospital transmission can have a profound impact on overall epidemic dynamics. If lockdowns are put in place for a fixed time period and then released in a stepwise manner (Fig. 4a–i), the total infected population in the community decreases from 27% in the high hospital transmission scenario to 12% and 7% in the intermediate and low transmission scenarios (Fig. 4g–i), with corresponding decreases in the percentages

of HCWs infected from 91% to 52% and 21% (Fig. 4d–f). Conversely, if instigation and release of lockdowns is driven by threshold infection rates in the community (Fig. 4j–u), the total number infected does not depend strongly on levels of hospital transmission (Fig. 4m–o), but the time spent in lockdown is reduced (Fig. 4p–r) and the efficiency with which lockdown averts infections compared with an unmitigated epidemic (Fig. 4s–u) is enhanced by reducing hospital transmission. These effects can be substantial despite the fact that, at any one time, the number of patients and HCWs is less than 2% of the total population.

## Discussion

Between 1% and 2% of hospital admissions are likely to have acquired SARS-CoV-2 infection while in hospital during the 'second wave' in England, with only a minority of these infections correctly classified as 'healthcare-associated' based purely on the time elapsed between admission and positive test. Investigation of the time series of hospital-acquired infections with a regression model indicated that patients who themselves acquired SARS-CoV-2 infection in hospital were the main drivers of transmission to patients, whereas transmission from both HCWs and nosocomially infected patients were of similar importance for transmission to HCWs (Fig. 3f,g). HCW vaccination was associated with large reductions in infection rates and there was evidence that aspects of hospital building design could modulate such transmission; in particular, a higher proportion of beds in single rooms was associated with decreased transmission risk, as was increased hospital building heated volume per bed, consistent with predictions from theoretical models for the spread of airborne infections in enclosed spaces[18].

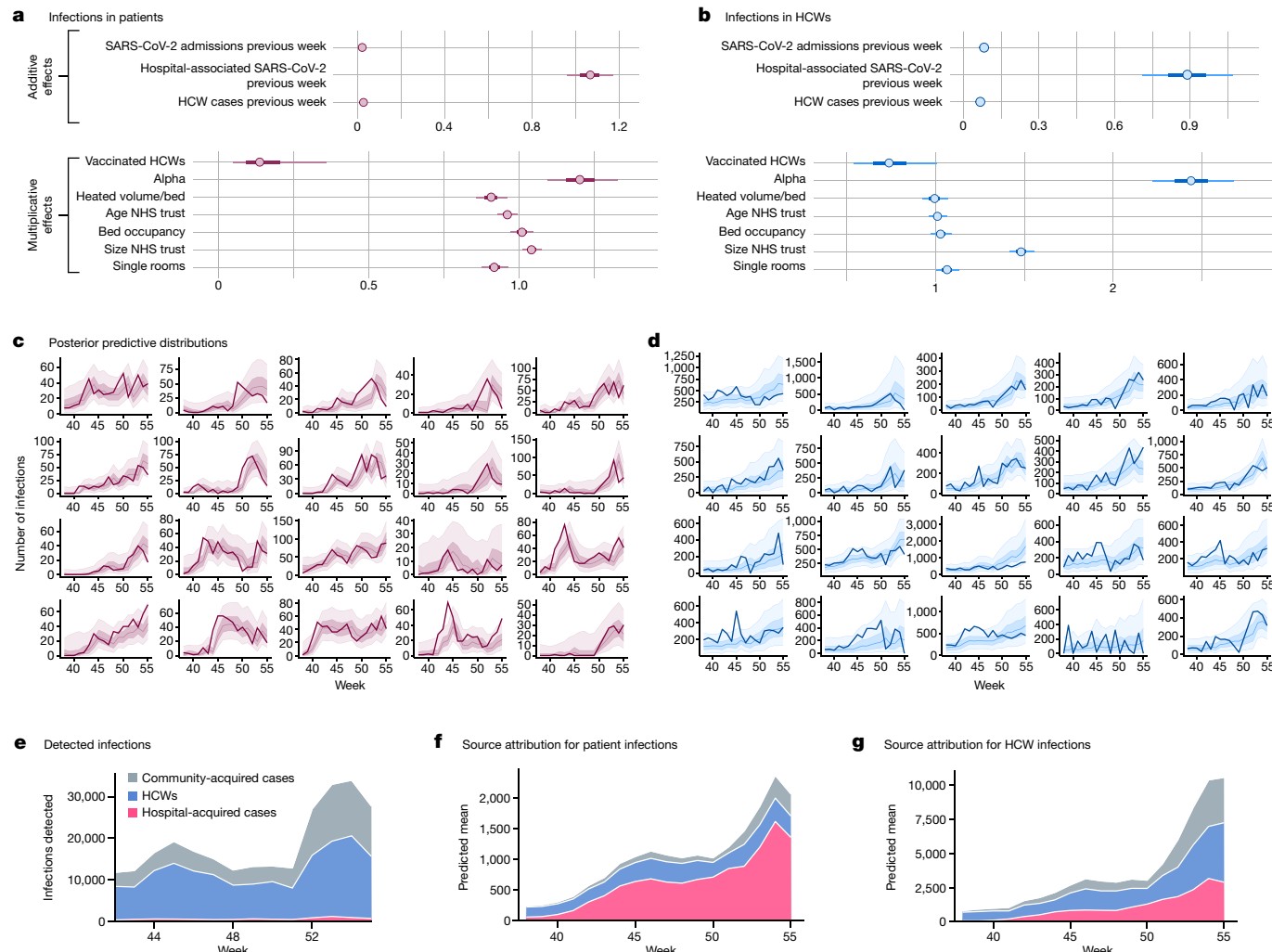

**Fig. 3 | Factors associated with healthcare-associated SARS-CoV-2 in patients, HCWs and predictive distributions. a,b,** Additive effects associated with categories of host infections and multiplicative effects of vaccine coverage in HCWs, Alpha prevalence and trust characteristics (posterior means and 50% and 90% CrIs are shown and results are on the basis of 1,661 hospital trust weeks of data from 96 different hospital trusts). For multiplicative effects, values below 1 indicate an association with reduced infection rates. Note that in the model for infections in patients (**a**), HCW vaccine coverage acts by modulating transmission associated with infected HCWs, whereas in the model for infections in HCWs (**b**) it has a global effect, modulating the overall rate of infection. **c,d,** Associated posterior predictive distributions for the number of detected infections by week in the 20 largest trusts are shown for infections in patients (**c**) and in infections in HCWs (**d**). Bold solid lines correspond to observed values, shaded regions correspond to 50% and 90% CrIs and the central lines within the shaded regions are median values from the posterior sample. **e–g,** For all trusts, classifications of detected infections by week (**e**) and contributions to predicted hospital-acquired infections in patients (**f**) and HCWs (**g**) from the three categories of infected hosts predicted by the full negative binomial regression models accounting for HCW vaccination and Alpha variant effects are shown. When the dependent variable is healthcare-associated SARS-CoV-2 infection in patients, these results use the ECDC definitions of definite and probable healthcare-associated infection (see Supplementary Information section 2 and the Supplementary Results for models using other definitions).

Although lack of genomic data means we cannot conclusively demonstrate transmission, our findings accord with focused local investigations with densely sampled viral genome sequences. Such studies indicate that many hospital-onset infections not meeting ECDC definitions for healthcare-associated infection are hospital-acquired and highlight the importance of superspreading[5,19]. Such superspreading is implicit in our negative binomial models, which attribute 80% of detected patient–patient transmission events from nosocomially infected patients to about 20% of infected patients (Extended Data Fig. 7). Also aligned with our findings are conclusions from local studies that hospital-acquired infection in patients was primarily due to transmission from nosocomially infected patients, whereas sources for HCW infections came from patients and HCWs in approximately equal proportions[9,19,20].

National infection prevention and control (IPC) guidance in England at the start of June 2020 emphasized respiratory and hand hygiene, use of face masks for patients and HCWs, cohorting of patients and staff, environmental decontamination, ventilation and staff social distancing. Screening of all patients for SARS-CoV-2 during the first 7 d of their hospital stay was recommended throughout the period, but some trusts went beyond these requirements by performing weekly testing. Records of such measures were not kept at a national level and lack of centrally collected data on trust-specific IPC measures means that effective interventions may have gone unrecognized and may potentially confound observed associations. Simulation studies, however, indicate that high-frequency asymptomatic screening and rapid isolation of patients with suspected SARS-CoV-2 can substantially reduce SARS-CoV-2 transmission in healthcare settings[21,22], and highlight the

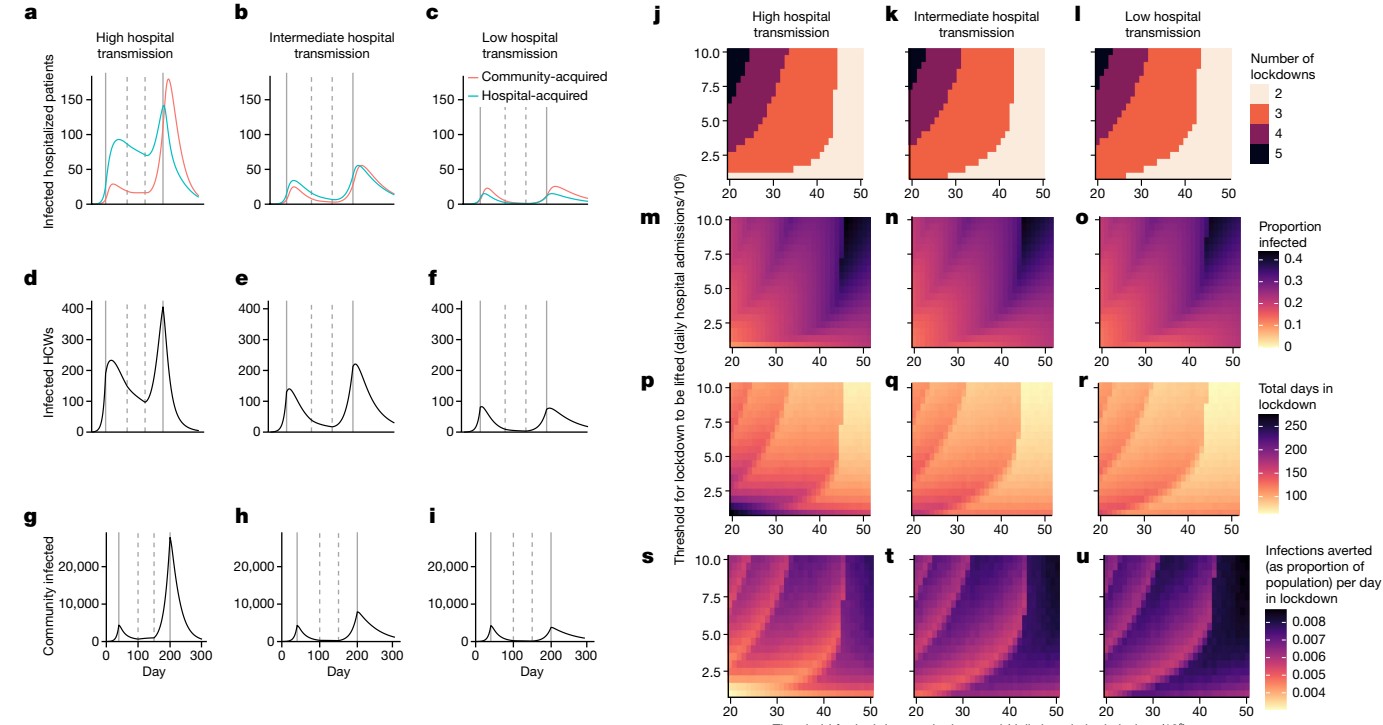

**Fig. 4 | Dynamics of community and hospital infections. a–i,** Results from simulation runs showing infected patients under high (**a**), intermediate (**b**) and low (**c**) rates of hospital transmission, in which rates of hospital transmission in intermediate and low scenarios are, respectively, 25% and 50% lower than the high hospital transmission scenario without altering parameters related to community transmission. Corresponding numbers of infected HCWs (**d**–**f**) and total numbers infected in the community (**g**–**i**) under the same scenarios are also shown. Assumed population sizes for community, hospital inpatients and HCWs are 500,000, 1,000 and 4,000, respectively. Solid vertical lines correspond to initiation of 'lockdown' measures, which are assumed to reduce person-to-person transmission rates in the community by 80% for the first lockdown and 70% for the second. The two broken vertical lines correspond to progressive release of lockdown measures, here assumed to result in transmission rates in the community that are reduced by 70% (after 100 d)

and 40% (after a further 50 d) compared with the pre-intervention rate. **j–u,** The same three hospital transmission scenarios are used when considering threshold-driven lockdown measures (**j**–**u**), when lockdown measures are initiated and released on the basis of per capita infection rates in the community being above or below prespecified thresholds. In these scenarios, when lockdown is in place person-to-person transmission rates in the community are assumed to be reduced by 90% compared with pre-intervention levels, whereas release of lockdown is followed by community transmission rates that are 50% of those before the first lockdown. Outputs shown are number of lockdowns under high (**j**), intermediate (**k**) and low (**l**) hospital transmission scenarios, the total proportion of the population infected (**m**–**o**), total days in lockdown (**p**–**r**), and infections averted per day in lockdown (**s**–**u**) under the same three scenarios.

importance of contact tracing[23]. Further limitations include the lack of PCR sensitivity estimates specific to the Alpha variant or conditioned on symptoms, and lack of consideration of vaccination in the patient population for which we lacked data. Although vaccine rollout to the over 70s and clinically extremely vulnerable began on 18 January 2021 in England, residents in care homes for older adults and their carers and those aged 80 and over were first eligible for vaccination on 8 December 2020; we estimate that 18% of those aged 80 and over and no more than 10% of those aged 70–79 may have had some degree of vaccine protection by the last week of the study (Supplementary Information section 2.4). We did not consider outpatients in this work as they are typically cared for in separate outpatient clinic settings distinct from the wards of acute hospitals.

The factors that make it hard to prevent SARS-CoV-2 transmission are relevant for hospitals everywhere. Although some well-resourced hospitals avoided large-scale nosocomial transmission in early 2020 (refs. 24–26), even in high-income settings the extent of such transmission showed considerable variation between hospitals[8]. Seroprevalence data before vaccination in HCWs also indicate a high degree of heterogeneity between hospitals even in the same countries and are consistent with high levels of nosocomial transmission in many settings (Extended Data Fig. 8). Hospitals in resource-limited settings face particular challenges due to poorly funded IPC activities, lack of capacity

to carry out routine testing, lack of isolation facilities and high levels of patient crowding, but attempts to systematically quantify the extent of such transmission outside high-income countries are currently lacking.

Our findings have implications for control policies. First, they highlight the importance of early identification and prompt initiation of control measures for patients with new hospital-acquired infections and for other patients they may have infected. Second, they reinforce the need for measures that reduce transmission from patients with asymptomatic infection in non-COVID-19 hospital areas, including improved ventilation, use of face coverings by patients and staff, increased distancing between beds, minimizing patient movements within and between wards and promotion of hand hygiene[27,28]. Third, our findings support efforts to prioritize HCWs for COVID-19 vaccination both due to direct protection to HCWs and due to indirect protection offered to patients. Fourth, the findings highlight the need to prioritize research into effective methods of reducing hospital transmission of airborne pathogens for which evidence is currently lacking[29], including ward design and air filtration systems[30]. Although our analysis focuses on nosocomial transmission early in the pandemic and before widespread vaccine coverage, the emergence of the highly contagious Omicron variants of SARS-CoV-2 has presented further infection control challenges, with high rates of hospital-onset infection reported despite high vaccine coverage, universal masking, admission

testing and symptom-based screening; anecdotal reports indicate that heightened control measures may be needed to suppress nosocomial spread[31].

Finally, our findings show that hospital transmission can have a substantial impact on epidemic dynamics in the wider community. In particular, the role of hospital transmission in seeding COVID-19 into care homes and other vulnerable groups in the community must be further investigated in light of the finding that much of the hospital transmission is likely to be unobserved.

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

# Methods

## Quantifying the number of hospital-acquired infections

**Inferential approach.** We estimate the total number of hospital-acquired infections in trust $i$ (combining observed and unobserved infections), $z_i$, by applying Bayes' formula:

$$P(z_i|y_i, \pi_i') = P(y_i|z_i, \pi_i')P(z_i)/P(y_i|\pi_i')$$

where $\pi_i'$, represents the probability that an infection acquired by a patient in trust $i$ is both detected by a PCR test and meets the definition of a hospital-acquired infection (which requires the first positive sample to be taken 15 or more days after the day the patient is admitted to the trust and before patient discharge), assumed independent of $z_i$. Here, $P(y_i|z_i, \pi_i')$ represents the binomial likelihood of observing $y_i$ identified hospital-acquired infections, $P(z_i)$ is the prior distribution for the total number of infections, which we take to be uniform (bounded by 0 and 20,000), and we calculate $P(y_i|\pi_i')$ using the law of total probability $P(y_i|\pi_i') = \sum_l P(y_i|\pi_i', z_i = l)P(z_i = l)$.

**Effect of testing policy.** The probability that a new hospital-acquired infection in trust $i$ is detected is given by $\pi_i = \sum_{m,d} \gamma_{imd}P_{imd}$, where $P_{imd}$ is the probability that a patient admitted to trust $i$ with length of stay $m$ and infected on day of stay $d$ (where $d \leq m$) has a positive PCR test while in hospital and $\gamma_{imd}$ is the probability that, given a new hospital-acquired infection in trust $i$ occurs, it occurs in a patient with length of stay $m$ on day of stay $d$. Similarly, the probability that a new hospital-acquired infection is both detected and meets the definition of a hospital-acquired infection is

$$\pi_i' = \sum_{m,d} \gamma_{imd}P_{imd}'$$

where $P_{imd}'$ is the probability that an infection in a patient admitted to trust $i$ with length of stay $m$ infected on day of stay $d$ is both detected and meets the definition of a hospital-acquired infection.

Consider an infection that a patient acquires $d$ days after the day the patient is admitted to the hospital. The testing policy in place in the trust during the patient's stay, the day of infection and the incubation period distribution together determine the probability that a patient is tested on day $k$ after the patient is infected (for $k = 0, 1, 2, 3 \ldots$). We assume the test has a specificity of 1. Let $\phi_k$ represent the sensitivity of a PCR test taken $k$ days after the date of infection, and let $\tau_{ik}$ represent the probability that such a test is performed $k$ days after the infection event, assumed to be independent for each value of $k$ of whether a test is performed on any other day. Then, $P_{imd} = 1 - \prod_{k=d\ldots m} (1 - \tau_{i(k-d)}\phi_{k-d})$.

The corresponding probability, $P_{imd}'$, is zero for $m < 15$ (because in that case the definition of hospital-acquired infection is not met); otherwise, it is given by the probability that there is no positive test before day 15 and at least one positive test after. For $d \geq 15$ this probability is identical to $P_{imd}$; otherwise, it is given by

$$P_{imd}' = \prod_{k=d\ldots14} (1 - \tau_{i(k-d)}\phi_{k-d})(1 - \prod_{k=15\ldots m} (1 - \tau_{i(k-d)}\phi_{k-d})).$$

If $\lambda_{im}$ represents the probability that a patient at risk of nosocomial infection with SARS-CoV-2 admitted to trust $i$ has a length of stay of $m$ days, then, on a given day, the expected proportion of patients who both have a length of stay of $m$ days and are currently on day of stay $d$ is given by $\psi_{imd} = \left[\frac{\lambda_{im}m}{\sum_n \lambda_{in}n}\right]\frac{1}{m}I(m \geq d)$, where $I(m \geq d)$ is the indicator function, $\left[\frac{\lambda_{im}m}{\sum_n \lambda_{in}n}\right]$ is the probability that on a randomly chosen day a randomly chosen patient has a length of stay $m$ and $\frac{1}{m}$ is the probability that this randomly chosen day is day $d$ of stay. Analysis of individual-level patient data indicates that although daily risk of infection changes over calendar time, it does not vary appreciably with day of stay $d$ for typical lengths of stays[9], and we therefore approximate $\gamma_{imd}$ by $\psi_{imd}$ which we estimate on the basis of the reported lengths of stays of completed episodes of patients admitted to each trust over the time period considered. This will represent a reasonable approximation provided that the infection hazard is small and roughly constant over a patient's hospital stay.

**Testing policies considered.** We consider several different testing policies, which determine the probability values that the test is performed on day $k$ after infection in trust $i$ ($\tau_{ik}$), as exact data on what policies were available in each trust are unavailable.

The minimal testing policy, which involves the fewest tests, requires only that patients displaying symptoms of COVID-19 are tested, and we assume all such patients are tested on a single occasion, the date of symptom onset. When this policy is in place, the time of testing of patients with hospital-acquired infections, in relation to the time of infection, is determined by the incubation period and such a test is assumed to be performed if and only if the patient develops symptoms on or before the day of discharge. A second testing policy extends this by assuming that in the event of a negative screening result from a patient with symptoms, daily testing will continue to be performed until patient discharge, the first positive test or three consecutive negative tests (whichever occurs first). We consider further testing policies which combine symptomatic testing (without retesting if negative) with routine asymptomatic testing. In these policies all patients who have not already tested positive are screened at predetermined intervals using the same PCR test. We consider weekly, twice weekly, three times weekly and daily testing of all in-patients as well as a policy of testing twice in the first week of stay (in accordance with national guidance in England).

**Accounting for uncertainty in test sensitivity, incubation period distribution and the proportion of infections that are symptomatic.** For a given length-of-stay distribution, incubation period distribution, PCR sensitivity profile and probability that infection is symptomatic, the calculations outlined above to determine the probability that an infection is detected or both detected and classified as a hospital-acquired infection are deterministic, and require no simulation. We account for uncertainty in these quantities through a Monte Carlo sampling scheme, at each iteration sampling new values for PCR sensitivities, the incubation period distribution and the proportion of infections that are symptomatic. For PCR sensitivities, we directly sample from the posterior distribution reported by Hellewell et al.[16]. For the incubation period we assume a lognormal distribution, and sample the parameters from normal distributions with means (s.d.) of 1.621 (0.064) and 0.418 (0.069) as estimated by Lauer et al.[32]. Estimates of the proportions of infections that are symptomatic are taken from Mizumoto et al.[33] and this quantity is sampled from a normal distribution with mean (s.d.) of 0.82 (0.012). Length-of-stay distributions are directly obtained from the Secondary Uses Service for NHS acute trusts, excluding: (1) patients who were admitted with PCR-confirmed COVID-19; (2) patients who had samples taken in the first 7 d of their hospital stay that were PCR positive for SARS-CoV-2; and (3) patients with a length of stay of less than 1 d. In the primary analysis we use aggregate length-of-stay data for all trusts taken from the 12 month period from 1 March 2020. We also present results from two sensitivity analyses: in the first we use trust-specific $\lambda_{im}$ values; in the second we allow for the possibility that length-of-stay distributions change over time and use period-specific empirical length-of-stay distributions from the periods: June to August 2020; September to November 2020; and December 2020 to February 2021.

**Quantifying drivers of nosocomial transmission.** We used generalized linear mixed models to quantify factors associated with nosocomial transmission. In these models the dependent variable was either the observed number of healthcare-associated infections in trust $i$ and week $j$ among patients, $y_{ij}$, or the imputed number of infections in HCWs, $y_{ij}'$. When the dependent variable was healthcare-associated

infections in patients, we used ECDC criteria, repeating the analysis using three different classifications of healthcare-associated infection: (1) definite; (2) definite and probable; (3) definite, probable and indeterminate. Three classes of independent variables were considered: (1) known exposures to others in the same trust infected with SARS-CoV-2 to account for within-trust temporal dependencies, with separate terms corresponding to exposures in the previous week to patients with community-onset SARS-CoV-2 infections ($z_{i(j-1)}$), patients with hospital-acquired SARS-CoV-2 ($y_{i(j-1)}$) and HCWs with SARS-CoV-2 ($y'_{i(j-1)}$); (2) characteristics of the trusts that were considered, a priori, to be plausibly linked to hospital transmission: bed occupancy, provision of single rooms, age of hospital buildings, heated hospital building air volume per bed and size (number of acute care beds); (3) regional data including vaccine coverage among HCWs and the proportion of isolates represented by the Alpha variant. Models were formulated to reflect presumed mechanisms generating the data, and we used negative binomial models with identity link functions, allowing the number of exposures to different categories of SARS-CoV-2 infections to contribute additively to the predicted number of weekly detected infections, while allowing for multiplicative effects of the other terms. In models for which the dependent variable represented hospital-acquired infections in patients, the HCW vaccination effect was assumed to act only through a multiplicative term affecting transmission related to exposures to HCWs. By contrast, when the dependent variable represented infections in HCWs, vaccine exposure was allowed to have a multiplicative effect on the overall expected number of infections. Formally, we define the full model for infections in patients in trust $i$ and week $j$ (which we refer to as model P1.1.1) as:

$$y_{ij} \sim \text{neg bin}(\mu_{ij}, \varphi_{ij}),$$

where $\mu_{ij}$ represents the mean and the variance is given by $\mu_{ij} + \mu_{ij}^2/\varphi_{ij}$.

In the full model $\mu_{ij} = (a_i + by_{i(j-1)} + c_{ij}y'_{i(j-1)} + dz_{i(j-1)})m_{ij}n_{ij}$

$m_{ij} = \exp(q \times \text{single rooms}_i + r \times \text{trust size}_i + s \times \text{occupancy}_{i(j-1)}$
$\qquad + t \times \text{trust age}_{ij} + u \times \text{trust volume per bed}_{ij})$

$n_{ij} = \exp(w \times \text{proportion Alpha variant}_{ij})$

$c_{ij} = c \times \exp(v \times \text{HCW vax}_{i(j-1)})$

$\varphi_{ij} = \varphi_0 + k_i y_{i(j-1)}.$

$a_i \sim N(a_0, \sigma_a^2)$

$k_i \sim N(k_0, \sigma_k^2).$

The expression for the dispersion parameter of the negative binomial distribution, $\varphi_{ij}$, reflects the fact that the sum of $n$ independent negative binomially distributed random variables with mean $\mu$ and dispersion parameter $\varphi$ will itself have a negative binomial distribution with mean $n\mu$ and dispersion parameter $n\varphi$. Thus, in the idealized case that each of $n$ nosocomially infected patients in 1 week has a fully observed negative binomially distributed offspring distribution the next week with mean $\mu$ and dispersion parameter $\varphi$, then the total number of nosocomial infections observed would have a negative binomial distribution with parameters $n\mu$ and $n\varphi$. The $a_i$ represents a trust-level random effect term to account for within-trust dependency. We also considered two nested models, P1.1.0 and P1.0.0, obtained by setting the terms $q$, $r$, $s$, $t$ and $u$ to zero in both cases (that is, removing the trust-level terms) and by additionally setting the terms $v$ and $w$ to zero in the latter case (that is, removing regional vaccine- and variant-related terms). As a further sensitivity analysis, we also considered a model that allowed for time-varying changes in the number of hospital-acquired infections not accounted for by the covariates, by setting

$$\mu_{ij} = (1 + s(j))(a_i + by_{i(j-1)} + c_{ij}y'_{i(j-1)} + dz_{i(j-1)})m_{ij}n_{ij}$$

where $s(j)$ is a degree 3 spline with 6 equally spaced knots. We refer to this model as P1.1.1.tv. Similar models were used when the dependent variable was HCW infections, except that the HCW vaccine effect was included in the multiplicative term $m_{ij}$ instead of operating only through the $c_{ij}$ term.

We used normal(0,1) prior distributions by default for model parameters, except for variance terms $\sigma_a^2$ and $\sigma_k^2$ for which we used half-Cauchy(0,1) prior distributions, and $\varphi$ for which a half-normal(0,1) prior distribution was specified for the transformed parameter $1/\sqrt{\varphi_0}$. All analyses were performed in Stan[34] using the rstan package v.2.21.1 in R (ref. 35), running each model with four chains using 1,000 iterations for warm-up and 5,000 iterations for sampling.

In the main analysis, we used weekly aggregated data, counting week numbers as 1 plus the number of complete 7 d periods since 1 January 2020. We included only acute hospital trusts in this analysis, and excluded trusts that predominantly admitted children.

**Imputation method for weekly number of infections in HCWs.** Situation reports included fields allowing quantification of nosocomial transmission and number of HCWs isolated due to COVID-19 from 5 June 2020, but analysis here is restricted to data from week 42 (beginning 14 October 2020) to week 55 (beginning 13 January 2021), reflecting the date range for which all fields used in the analysis were consistently reported. Because situation reports did not explicitly include data on the number of infections in HCWs, only the number of HCWs absent due to COVID-19 on each day, we imputed the weekly number of infections among HCWs at each trust. We did this by first subtracting from the number of reported HCW COVID-19 absences in each trust on each day the reported number of such absences due to contact tracing and isolation policies (reflecting likely COVID-19 exposures in the community) to give $a_t$, the number of HCWs absent on day $t$ due to COVID-19 infection potentially arising from occupational exposure. Then, assuming that each HCW with COVID-19 was isolated for 10 d and assuming that durations of these absences were initially uniformly distributed (starting from week 36), the number imputed to have entered isolation on day $t$, $x_t$, was taken as $x_t = a_{t+1} + x_{t-10} - a_t$. For each trust we performed these calculations ten times, sampling the initial duration of staff absences from a multinomial distribution assigning equal probabilities to durations of 1 … 10 d, and then took the average (rounded to the nearest integer) of these samples. In some trusts it was evident that some days with missing HCW isolation data had been coded as zeroes. When such zeroes fell between daily counts in excess of ten we treated them as missing data and replaced them with the last number carried forward. Any negative numbers for daily imputed HCW infections resulting from the above procedure were replaced with zeroes.

Although data on healthcare-associated infections in patients were recorded consistently by all trusts throughout the inclusion period, in some trusts data on HCW absences due to COVID-19 were missing or had been recorded inconsistently throughout the inclusion period. Excluding such trusts and those with missing data for independent variables left 96 of the original 145 trusts included in the analysis.

## Negative control outcomes

We used as a negative outcome control the number of patients admitted with community-acquired SARS-CoV-2 infection as the outcome variable. We performed three analyses in which we adopted this negative control as our dependent variable, corresponding to models P1.1.1, P1.1.0 and P1.0.0 as defined above.

## Hospital–community interaction model

We modelled hospital–community interaction using ordinary differential equations for an expanded susceptible/exposed/infectious/removed model (Extended Data Fig. 9). This model included

separate compartments for people in the community ($S_C$, $E1_C$, $E2_C$, $I1_C$, $I2_C$, $I'_C$, $R_C$), patients in hospital ($S_H$, $E1_H$, $E2_H$, $I1_H$, $I2_H$, $I'_H$, $R_H$) and HCWs ($S_{HCW}$, $E1_{HCW}$, $E2_{HCW}$, $I1_{HCW}$, $I2_{HCW}$, $I'_{HCW}$, $R_{HCW}$), in which the two exposed compartments ($E1$ and $E2$) and the two infectious compartments ($I1$ and $I2$) for each subpopulation correspond to assumptions of an Erlang-distributed latent and infectious period with shape parameter 2, whereas the $I'$ compartments represent people with severe disease potentially requiring hospitalization. The model allowed for patient–patient, HCW–HCW, HCW–patient and community–HCW transmission, as well as movements of people between the community and hospital. In the interest of simplicity, we neglect hospitalization of HCWs who account for about 1% of the total population.

We used the model to explore the impact of hospital transmission on overall epidemic dynamics with the aim of providing qualitative insights. We considered outcomes from high, intermediate and low hospital transmission scenarios in which the primary epidemic control measure was restricting rates of contact in the community ('lockdowns'). This community control measure was assumed to have no direct impact on contact rates within hospitals as hospital infection control measures were in force throughout the study period irrespective of efforts aiming to limit community transmission. Full model details are provided in Supplementary Information section 1.2 and Supplementary Tables 1 and 2.

### Ethics approval

The study did not involve the collection of new patient data, or use any personal identifiable information, but used a combination of anonymized national aggregate data sources including C19SR01–COVID-19 Daily NHS Provider SitRep, and regionally aggregated vaccine coverage data from the SARS-CoV-2 immunity and reinfection evaluation (SIREN) study for which the study protocol was approved by the Berkshire Research Ethics Committee on 22 May 2020 with the vaccine amendment approved on 23 December 2020.

### Reporting summary

Further information on research design is available in the Nature Portfolio Reporting Summary linked to this article.

### Data availability

The data that support the findings of this study are available as described below. Infection data used for this analysis were taken from daily situation reports between 10 June 2020 and 17 February 2021 and shared privately with the Scientific Pandemic Influenza Group on Modelling (SPI-M). The start date was chosen as the first date that healthcare-associated infections were consistently reported across trusts, and the end date was taken to be 1 month after the start of vaccine rollout to the over 70s and clinically extremely vulnerable (18 January 2021). COVID-19 admission data for NHS trusts are publicly available by direct download from https://www.england. nhs.uk/statistics/statistical-work-areas/covid-19-hospital-activity/. Requests for data on healthcare-associated infections should be sent to J.V.R. (julie.robotham@phe.gov.uk) who will liaise with NHS England to provide access to these data and will respond to requests within 1 month. Trust-specific data used in the analysis not related to infections (number of single rooms, size, age, heated volume and bed occupancy) were derived from the Estates Returns Information Collection from NHS Digital (available for download at https://digital.nhs.uk/data-and-information/publications/statistical/estates-returns-information-collection), including only the following site types: general acute hospital, community hospital (with inpatient beds), mixed service hospital, specialist hospital (acute only). The number of single rooms was expressed as the number of beds in single rooms in the trust (including single bedrooms for patients with and without en-suite facilities and isolation rooms) divided by the number

of general and acute beds reported as being available in the trust in the last quarter of 2020. Hospital size was taken as the number of hospital beds available in the trust. A hospital building age score was taken as a weighted average of the proportion of floor area across hospital sites that was built before 1965, for which weights were taken as the building floor area. Data relating to vaccine coverage in HCWs were collected as part of the SIREN study (ISRCTN No. ISRCTN11041050)[36]. Data from this study are available on reasonable request and will be available through the Health Data Research UK CO-CONNECT platform and available for secondary analysis once the SIREN study has completed reporting. Using these data, we classified HCWs as being immunized if they had received at least one vaccine dose 3 or more weeks previously. Otherwise, they were considered un-immunized. SARS-CoV-2 variant data consisted of the proportion of characterized isolates that were attributed to the Alpha variant in each week for each NHS region. The prevalence of the Alpha variant by region and over time was determined by the proportion of tests with S-gene target failure status from PCR tests provided by Public Health England (accessed at https://github. com/epiforecasts/covid19.sgene.utla.rt)[37]. Patient length-of-stay data were taken from the Secondary Uses Service (SUS)[38]. Data to reconstruct the PCR sensitivity profile are available from https://github. com/cmmid/pcr-profile.

### Code availability

All code for the data analysis and simulations in this paper is available from https://zenodo.org/record/8123987 (ref. 39). Code to reconstruct the PCR sensitivity profile is available from https://github.com/cmmid/pcr-profile.

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

**Acknowledgements** We thank S. Hopkins and the SIREN Study team for permission to use data on vaccination coverage of HCWs, with particular thanks to S. Foulkes and E. Wellington who were instrumental in setting up linkage between SIREN and National Immunisation Management System (NIMS) records. We also thank C. Noakes for discussions. We acknowledge support from the National Institute for Health Research (NIHR) and UK Research and Innovation (grant no. COV0357; grant no. MR/V028456/1), the NIHR Health Protection Research Unit (HPRU) in Healthcare Associated Infections and Antimicrobial Resistance at Oxford University in partnership with the UK Health Security Agency (UKHSA) (grant no. NIHR200915), the NIHR Biomedical Research Centre, Oxford, and the NIHR HPRU in Emerging and Zoonotic Infections at the University of Liverpool in partnership with the UKHSA, in collaboration with Liverpool School of Tropical Medicine and the University of Oxford (grant no. NIHR200907). J.M.R. acknowledges support from the Medical Research Council (MRC) (grant no. MR/V038613/1). B.S.C. acknowledges support from the NIHR (grant no. PR-OD-1017-20006). T.M.P. was supported by the Society for Laboratory Automation and Screening (grant no. SLAS_VS2020). M.Y. is supported by the Singapore National Medical Research Council Research Fellowship (grant no. NMRC/Fellowship/0051/2017). C.A.D. acknowledges funding from the MRC Centre for Global Infectious Disease Analysis (reference no. MR/R015600/1), jointly funded by the UK MRC and the UK Foreign, Commonwealth & Development Office (FCDO), under the MRC/FCDO Concordat agreement, and is also part of the EDCTP2 programme supported by the European Union. S.F. is supported by the Wellcome Trust (grant no. 210758/Z/18/Z).

**Author contributions** B.S.C., J.V.R., G.M.K., S.E., T.M.P. and D.W.E. conceptualized this work. B.S.C. performed the statistical analysis. B.S.C., S.E. and Y.J. developed the dynamic model. S.E., Y.J., C.L., D.P., V.H., J.S., S.F., J.V.R. and G.M.K. obtained, processed and verified the underlying data. B.S.C. drafted the first version of the manuscript. All authors contributed to

interpretation of data and reviewed and edited subsequent versions of the manuscript. The corresponding author attests that all listed authors meet authorship criteria and that no others meeting the criteria have been omitted. The corresponding author accepts full responsibility for the work and/or the conduct of the study, had access to the data and controlled the decision to publish.

**Competing interests** D.W.E. declares personal fees from Gilead outside the submitted work. The other authors declare no competing interests.

**Additional information**
**Correspondence and requests for materials** should be addressed to Ben S. Cooper.

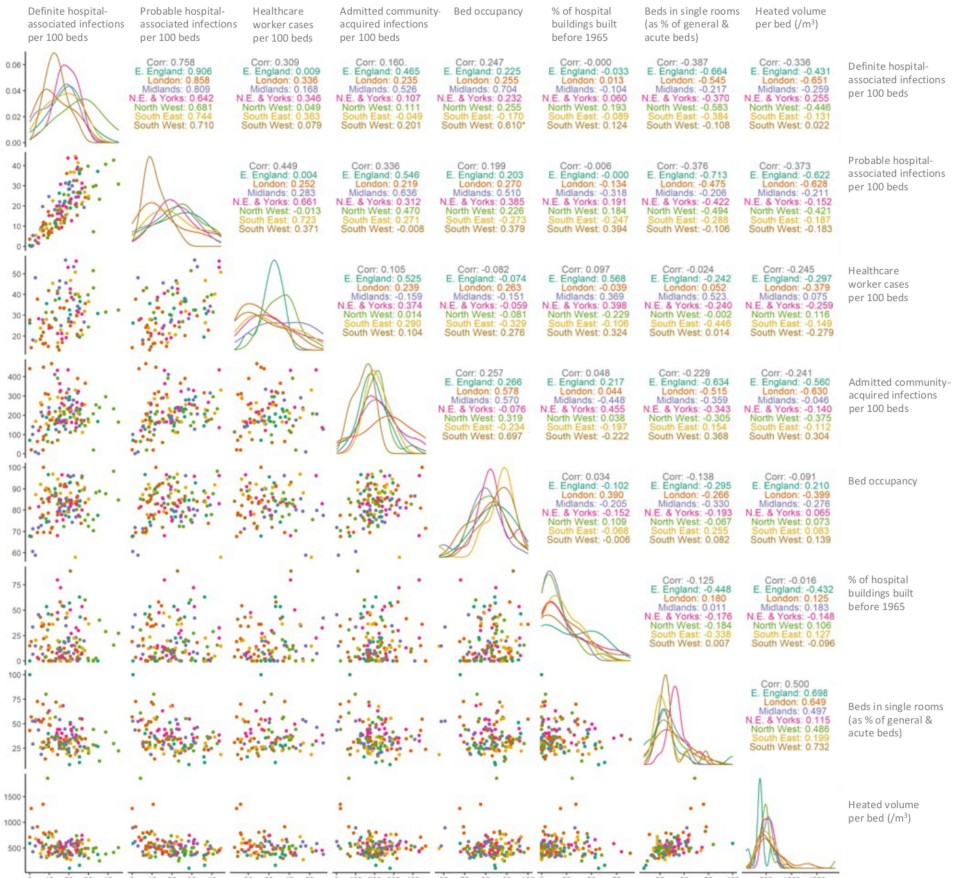

**Extended Data Fig. 1 | Pairs plot showing the relationships between cumulative trust-level infection rates and trust characteristics.** Diagonal elements show kernel density estimates for cumulative covid infections in trusts from 10th June 2020 to 17th February 2021: 1) definite hospital-acquired infections per 100 beds (defined as those first PCR positive 15 or more days after hospital admission); 2) probable hospital-acquired infections per 100 beds (those first PCR positive from 8–14 days after admission); 3) imputed healthcare worker (HCW) SARS-CoV-2 infections per 100 HCWs; 4) SARS-CoV-2 infections in hospitalised patients with community onset per 100 beds; 5) bed occupancy; 6) age of acute hospital buildings in the trust expressed as a weighted average of the percentage of hospital buildings constructed in 1964 or earlier, where weights are the hospital gross internal floor areas; 7) number of single room beds per trust (including isolation rooms) as a percentage of the number of general and acute beds available in the last quarter of 2020; 8) heated volume per bed ($m^3$). Below-diagonal elements show scatterplots, where each point (coloured according to NHS region) corresponds to a single NHS trust. Above diagonal elements show the Pearson correlation coefficients between pairs of variables, both nationally (in grey) and within each NHS region.

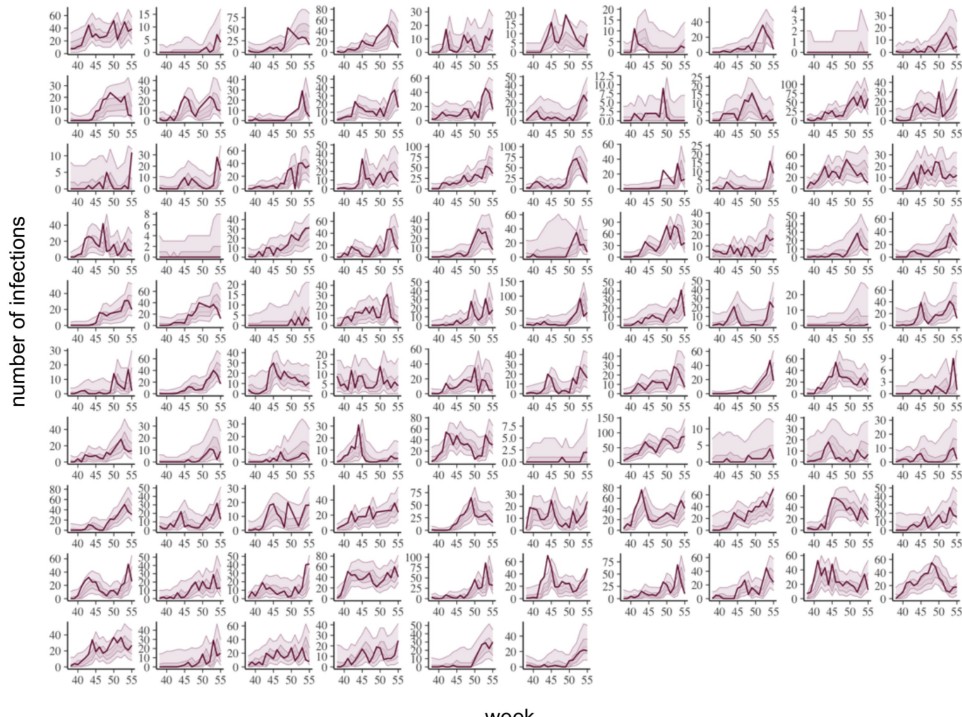

**Extended Data Fig. 2 | Infections in patients.** Posterior predictive distributions for all 96 trusts included in the analysis from model P1.1.1 where the outcome is probable and definite healthcare-associated infection. Bold solid lines correspond to observed values and shaded regions correspond to 50% and 90% CrIs.

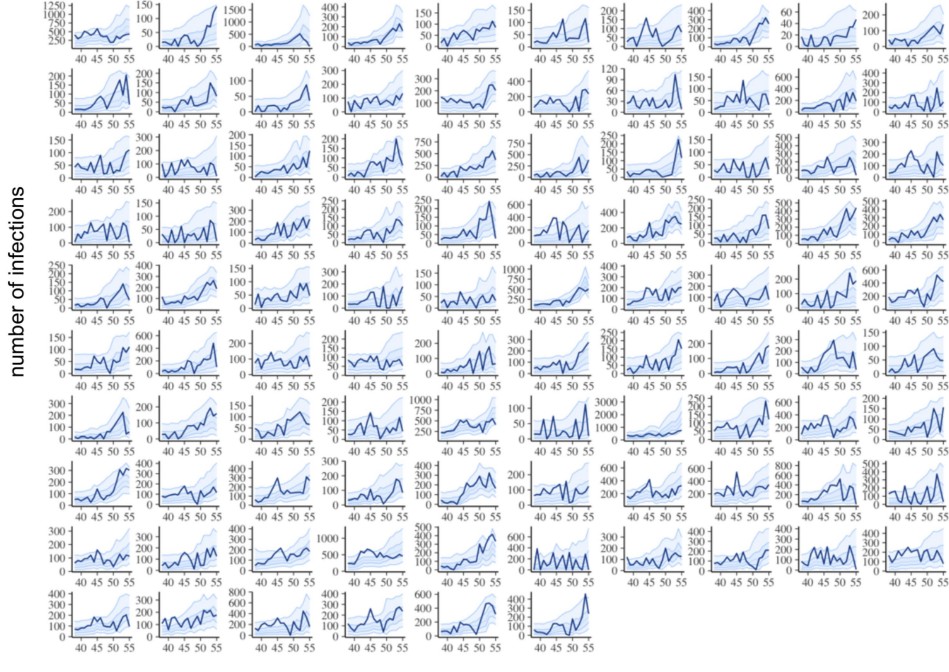

**Extended Data Fig. 3 | Infections in healthcare workers.** Posterior predictive distributions for all 96 trusts included in the analysis from model P1.1.1 where the outcome is infections in HCWs. Bold solid lines correspond to observed values and shaded regions correspond to 50% and 90% CrIs.

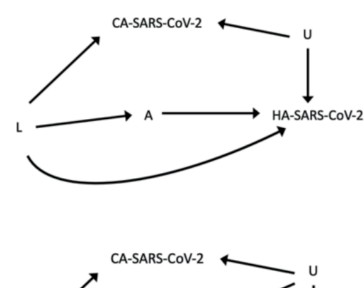

**a**

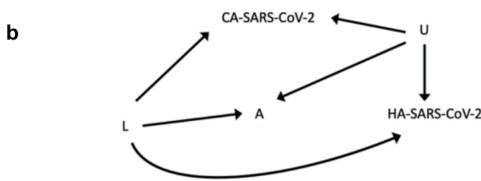

**b**

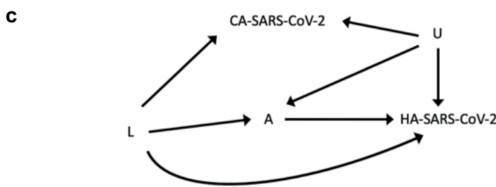

**c**

**Extended Data Fig. 4 | Directed acyclic graphs showing community-acquired SARS-CoV-2 (CA-SARS-CoV-2) infection as a negative control outcome for use in evaluating the relationship between an exposure, *A*, and hospital-acquired SARS-CoV-2 (HA-SARS-CoV-2).** Measured confounders, *L*, are assumed to be adjusted for in the analysis, while unmeasured variables, *U*, may distort the estimated measure of association between exposure and hospital-acquired SARS-CoV-2 infection, generating a non-causal association. (a) Suppose that exposure, *A*, is a cause of HA-SARS-CoV-2 but not of CA-SARS-CoV-2, while unmeasured variables, *U*, are causes of both HA-SARS-CoV-2 and CA-SARS-CoV-2 but not of *A* (for example, factors affecting susceptibility to infection). In this case, in an analysis that adjusts for *L*, the association between *A* and HA-SARS-CoV-2 is a consequence of the causal link between *A* and HA-SARS-CoV-2, and no such association would be seen between *A* and the control outcome, CA-SARS-CoV-2. b) Conversely, if *U* is a cause of *A*, HA-SARS-CoV-2 and CA-SARS-CoV-2, but *A* is neither a cause of HA-SARS-CoV-2 nor of CA-SARS-CoV-2 then in an analysis adjusting for *L* associations between *A* and HA-SARS-CoV-2 and between *A* and CA-SARS-CoV-2 are expected as a consequence of the confounding factors, *U*. If a) and b) were the only possible causal relationships to be considered, an association between *A* and HA-SARS-CoV-2 but not between *A* and CA-SARS-CoV-2 after adjusting for *L* would provide evidence in support of a), where *A* is a cause of HA-SARS-CoV-2, while an association between *A* and CA-SARS-CoV-2 (after adjusting for *L*), would support b) as the backdoor path through *U* is open. c) If *A* is both a cause of HA-SARS-CoV-2 and there are unmeasured confounders, *U*, an association between *A* and HA-SARS-CoV-2 after adjusting for *L* is a consequence of both the direct causal link and confounding; in this case we would also expect an association between *A* and CA-SARS-CoV-2 after adjusting for *L* arising entirely as a result of confounding.

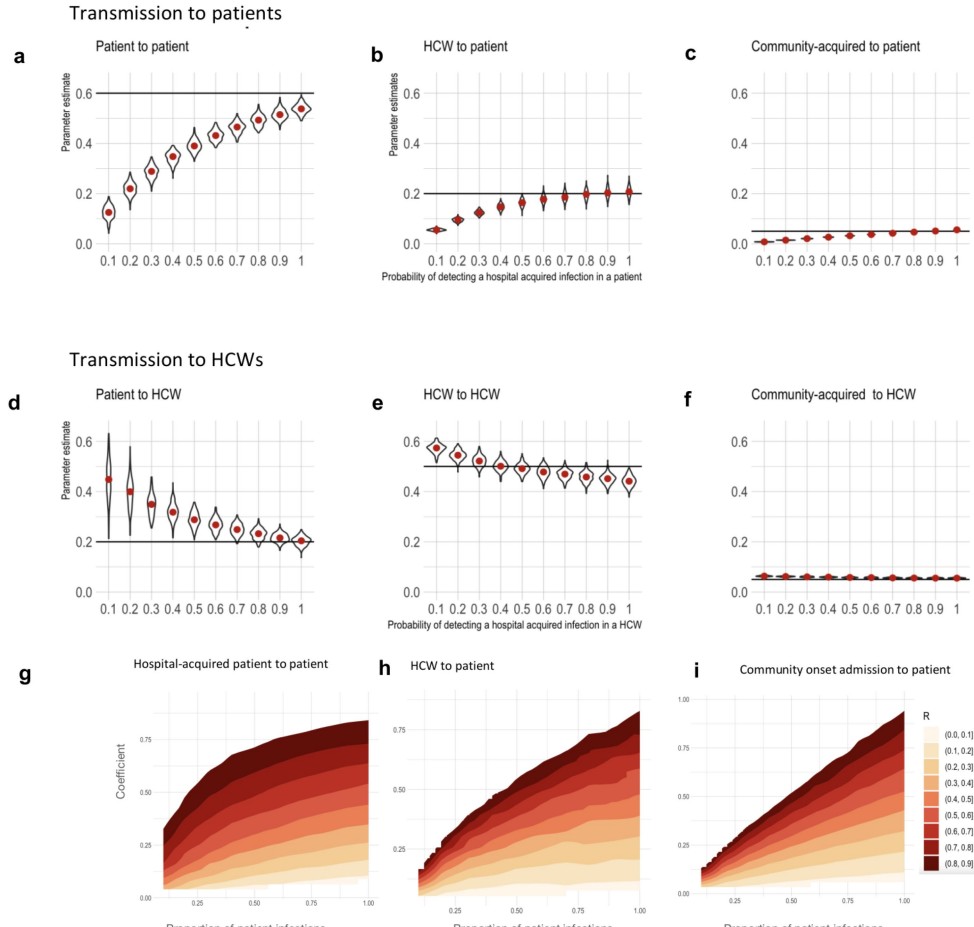

**Extended Data Fig. 5 | Results of a simulation study.** Parameter estimates from fitting a negative binomial auto-regression model to simulated data under different probabilities for observing hospital-acquired infections in patients (**a**-**f**). The thick horizontal line indicates the component of the reproduction number used when simulating data (for example, in (**a**) each patient with a hospital-acquired infection infects, on average, 0.6 other hospitalised patients). Red dots indicate the median from 100 simulations and the width in the violin plots is proportional to the density. Heatmaps (**g**-**i**) show how estimated model parameters from a negative binomial auto-regression model (y-axis) map onto reproduction numbers (shown by the colour scale) for different proportions of hospital-acquired infections observed in patients (x-axis). Reproduction numbers correspond to expected numbers of secondary infections in patients from patients who themselves became infected in hospital (**g**), secondary infections in patients from healthcare workers (**h**) and secondary infections in patients from patients admitted to hospital with COVID-19 (**i**).

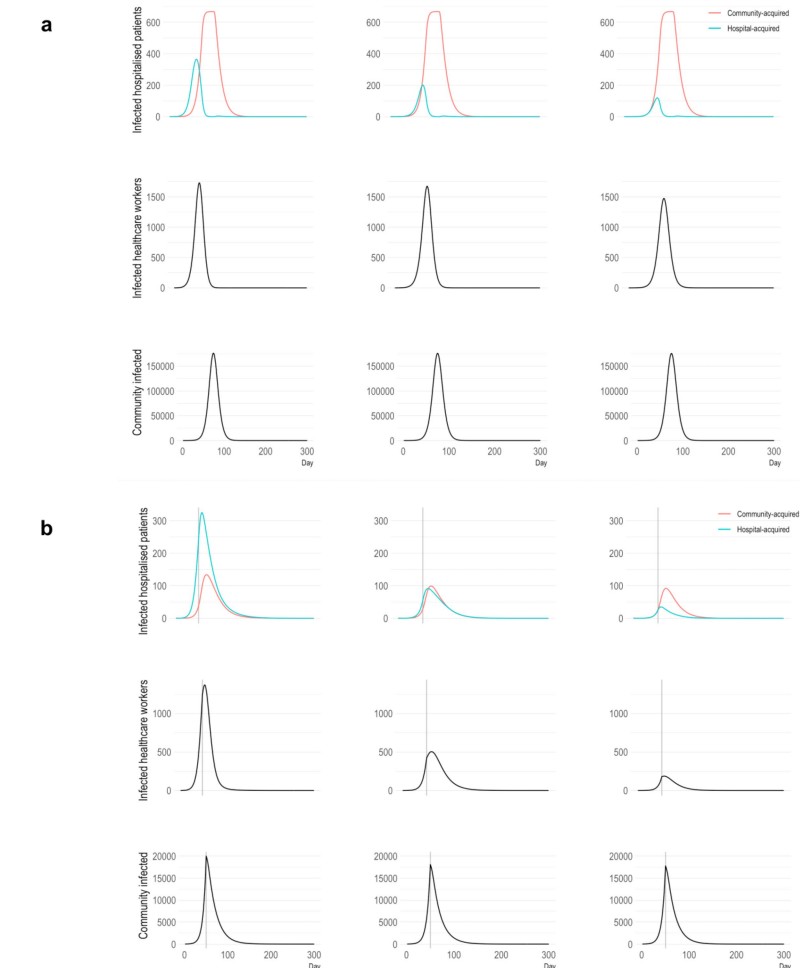

**Extended Data Fig. 6 | Additional output from deterministic model.** Dynamics of unmitigated epidemics under scenarios of high, intermediate and low transmission in hospitals (**a**). Dynamics of epidemics under scenarios of high, intermediate and low transmission in hospitals when a single "lockdown" intervention is introduced on day 50 (grey vertical line), which has the effect of stopping 90% of community-based transmission but no effect on hospital-based transmission (**b**).

**a**

**Probable and definite healthcare associated infection**

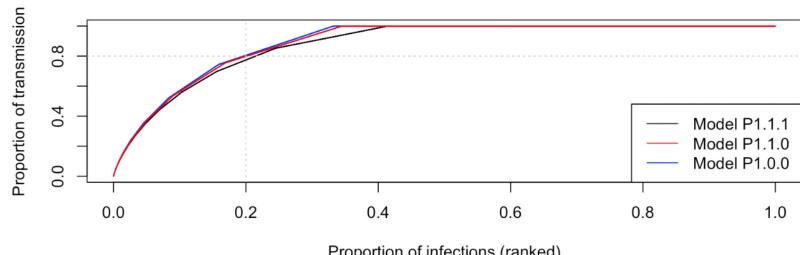

**b**

**Definite healthcare associated infection**

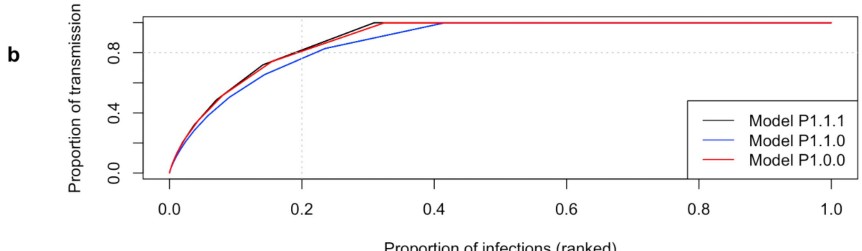

**Extended Data Fig. 7 | Proportion of all transmission due to a given proportion of infectious cases, where cases are ranked by infectiousness.** Results are obtained by simulation with $10^6$ samples using point estimates from models P1.1.1, P1.1.0 and P1.0.0 where the dependent variable is the number of probable and definite healthcare associated infections (**a**), and definite healthcare associated infections (**b**).

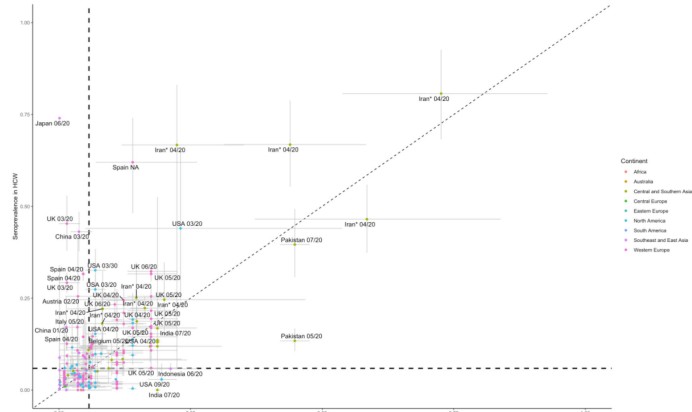

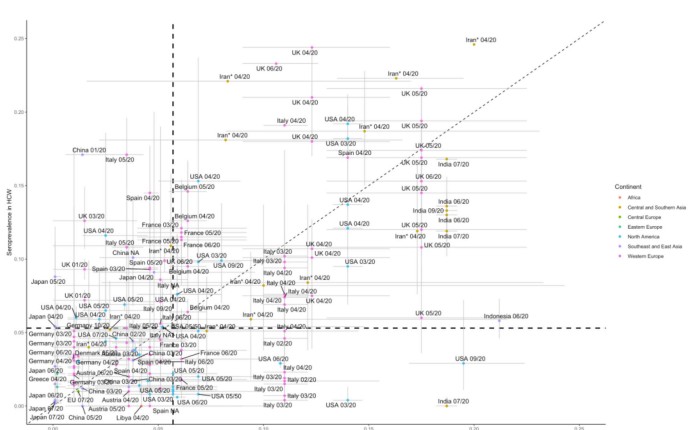

**Extended Data Fig. 8 | Seroprevalence in HCWs against seroprevalence in the community reported in the papers published before 16 May 2021 (means and 95% CIs).** Dashed horizontal and vertical lines are the reported median values of seroprevalence in HCWs and in the community, respectively. The dots are coloured by the continent in which the survey was performed. The label for each dot shows country and survey period (i.e. 01/20 means January 2020). *The study from Iran surveyed 18 cities and classified the survey populations into high-risk populations (including HCWs, pharmacy employees, taxis drivers, cashiers of supermarket chains, and bank employees) and general populations in the same city over the same survey period. The bottom panel plot shows a zoomed in part of the top panel.

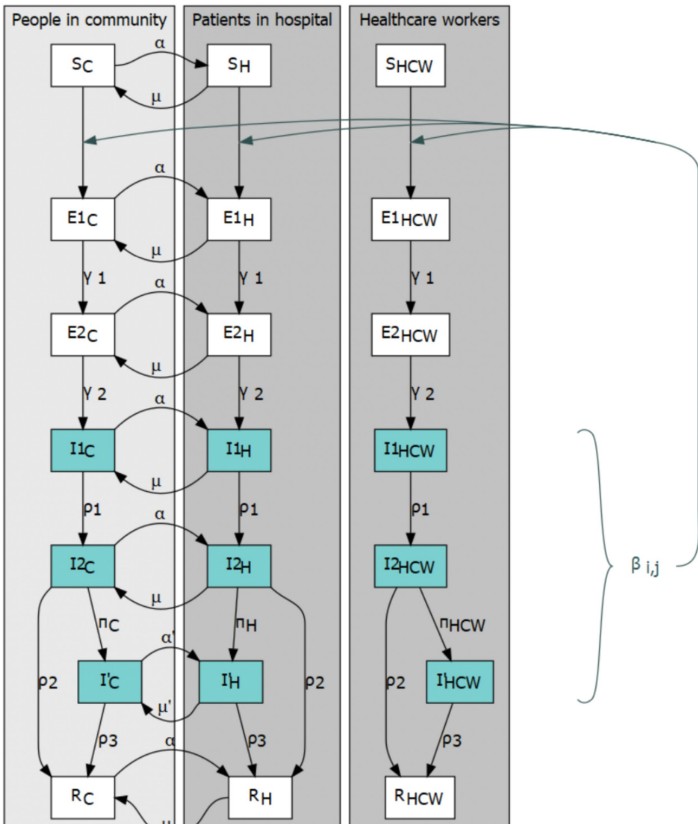

**Extended Data Fig. 9 | Flow diagram for the compartmental model coupling hospital and community dynamics.** Rectangles indicate infection states (S – susceptible to infection, E1 and E2 – infected but not yet infectious; I1 and I2 – infected and infectious; I' severe disease). These compartments are duplicated for people in the community (subscript C, left panel), patients in hospital (subscript H, centre panel) and healthcare workers (subscript HCW, right panel). Arrows indicate permitted movements between these states and Greek letters correspond to parameters controlling the rate of these movements. The two exposed pre-infectious states (E1, E2) and the two infectious states (I1, I2), are used to represent Erlang-distributed latent and infectious periods.

# Reporting Summary

## Statistics

For all statistical analyses, confirm that the following items are present in the figure legend, table legend, main text, or Methods section.

| n/a | Confirmed | |
|---|---|---|
| ☐ | ☒ | The exact sample size (*n*) for each experimental group/condition, given as a discrete number and unit of measurement |
| ☐ | ☒ | A statement on whether measurements were taken from distinct samples or whether the same sample was measured repeatedly |
| ☒ | ☐ | The statistical test(s) used AND whether they are one- or two-sided<br>*Only common tests should be described solely by name; describe more complex techniques in the Methods section.* |
| ☐ | ☒ | A description of all covariates tested |
| ☐ | ☒ | A description of any assumptions or corrections, such as tests of normality and adjustment for multiple comparisons |
| ☐ | ☒ | A full description of the statistical parameters including central tendency (e.g. means) or other basic estimates (e.g. regression coefficient) AND variation (e.g. standard deviation) or associated estimates of uncertainty (e.g. confidence intervals) |
| ☒ | ☐ | For null hypothesis testing, the test statistic (e.g. $F$, $t$, $r$) with confidence intervals, effect sizes, degrees of freedom and $P$ value noted<br>*Give P values as exact values whenever suitable.* |
| ☐ | ☒ | For Bayesian analysis, information on the choice of priors and Markov chain Monte Carlo settings |
| ☐ | ☒ | For hierarchical and complex designs, identification of the appropriate level for tests and full reporting of outcomes |
| ☐ | ☒ | Estimates of effect sizes (e.g. Cohen's *d*, Pearson's *r*), indicating how they were calculated |

*Our web collection on statistics for biologists contains articles on many of the points above.*

## Software and code

Policy information about availability of computer code

| | |
|---|---|
| Data collection | All data used in this study were either derived from publicly available data sources or shared with us privately through the Scientific Pandemic Influenza Group on Modelling Operational sub-group (SPI-M-O) and information on software used to collect data was not provided. |
| Data analysis | All analysis was performed using R version 4.2.0 using the rstan package version 2.21.1 for the regression models. Code written for this analysis is available at the following URL: https://zenodo.org/record/8123987. Code to reconstruct the PCR sensitivity profile is available from https://github.com/cmmid/pcr-profile. |

For manuscripts utilizing custom algorithms or software that are central to the research but not yet described in published literature, software must be made available to editors and reviewers. We strongly encourage code deposition in a community repository (e.g. GitHub). See the Nature Portfolio guidelines for submitting code & software for further information.

## Data

Policy information about availability of data

All manuscripts must include a data availability statement. This statement should provide the following information, where applicable:
- Accession codes, unique identifiers, or web links for publicly available datasets
- A description of any restrictions on data availability
- For clinical datasets or third party data, please ensure that the statement adheres to our policy

The data that support the findings of this study are available as described below. Infection data used for this analysis were taken from daily situation reports

between 10th June 2020 and 17th February 2021 and shared privately with the Scientific Pandemic Influenza Group on Modelling Operational sub-group (SPI-M-O). The start date was chosen as the first date that healthcare-associated infections were consistently reported across trusts, and the end date was taken to be one month after the start of vaccine roll-out to the over 70s and clinically extremely vulnerable (18th January 2021). COVID-19 admission data for NHS trusts are publicly available by direct download from https://www.england.nhs.uk/statistics/statistical-work-areas/covid-19-hospital-activity/. Requests for data on healthcare associated infections should be sent to Dr Julie Robotham (julie.robotham@phe.gov.uk) who will liaise with NHS England to provide access to these data and will respond to requests within one month. Trust-specific data used in the analysis not related to infections (number of single rooms, size, age, heated volume and bed occupancy) were derived from the Estates Returns Information Collection from NHS Digital (available for download at https://digital.nhs.uk/data-and-information/publications/statistical/estates-returns-information-collection) including only the following site types: general acute hospital, community hospital (with inpatient beds), mixed service hospital, specialist hospital (acute only). The number of single rooms was expressed as the number of beds in single rooms in the trust (including single bedrooms for patients with and without en-suite facilities and isolation rooms) divided by the number of general and acute beds reported as being available in the trust in the last quarter of 2020. Hospital size was taken as the number of hospital beds available in the trust. A hospital building age score was taken as a weighted average of the proportion of floor area across hospital sites that was built before 1965, where weights were taken as the building floor area.

Data relating to vaccine coverage in healthcare workers were collected as part of the SIREN study (ISRCTN Number. ISRCTN11041050). Data from this study are available on reasonable request to Dr Julie Robotham and will be available through the Health Data Research UK CO-CONNECT platform and available for secondary analysis once the SIREN study has completed reporting. Using these data we classified healthcare workers as being immunised if they had received at least one vaccine dose three or more weeks previously. Otherwise they were considered un-immunised. SARS-CoV-2 variant data consisted of the proportion of characterised isolates that were attributed to the Alpha variant in each week for each NHS region. The prevalence of the Alpha variant by region and over time was determined by the proportion of tests with S-gene target failure status from PCR tests provided by Public Health England accessed at (https://github.com/epiforecasts/covid19.sgene.utla.rt)36. Patient length of stay data were taken from Secondary Uses Service (SUS)37. Data to reconstruct the PCR sensitivity profile are available from https://github.com/cmmid/pcr-profile.

# Human research participants

Policy information about studies involving human research participants and Sex and Gender in Research.

| | |
|---|---|
| Reporting on sex and gender | Neither sex nor gender were considered in the analysis and we did not have access to data relating to sex or gender of those infected or at risk of infection. |
| Population characteristics | We did not consider age or other individual characteristics of patients |
| Recruitment | The study was a retrospective analysis of national datasets for England, so patients were not recruited into this study. |
| Ethics oversight | The study did not involve the collection of new patient data, or use any personal identifiable information, but used a combination of anonymised national aggregate data sources including C19SR01 - COVID-19 Daily NHS Provider SitRep, and regionally aggregated vaccine coverage data from the SIREN study for which the study protocol was approved by the Berkshire Research Ethics Committee on May 22, 2020 with the vaccine amendment approved on Dec 23, 2020. |

Note that full information on the approval of the study protocol must also be provided in the manuscript.

# Field-specific reporting

Please select the one below that is the best fit for your research. If you are not sure, read the appropriate sections before making your selection.

☒ Life sciences          ☐ Behavioural & social sciences          ☐ Ecological, evolutionary & environmental sciences

For a reference copy of the document with all sections, see nature.com/documents/nr-reporting-summary-flat.pdf

# Life sciences study design

All studies must disclose on these points even when the disclosure is negative.

| | |
|---|---|
| Sample size | This was a retrospective analysis of complete national data rather than a sample, and all adult NHS hospital Trusts in England with available data were included. This represented over 98% of the total NHS general and acute care bed capacity in England in 2020. . |
| Data exclusions | Date ranges were chosen to start when healthcare associated infection were first consistently reported in England, and to end before widespread vaccination would complicate interpretation of results. Trusts that exclusively cared for children were excluded because of the distinct epidemiology of SARS-CoV-2 in children. The regression analysis was limited to the 96 of the 145 NHS acute hospital trusts where the more detailed data required for this analysis was available. All exclusion criteria were pre-established. |
| Replication | This was a retrospective analysis of national data from England. Code is provided to enable the analysis to be replicated using other national data, but we did not perform such replication as data were not available to us. To help interpret estimated regression coefficients we performed a series of simulation studies, generating synthetic transmission data-sets from a multitype branching process model, applying an observation model to obtain partially observed infection data, and replicating the above analysis. |
| Randomization | Randomization was not applicable as this was a retrospective analysis of national data and exposures of interest were not assigned to |

| | |
|---|---|
| Randomization | hospitals by the investigators, precluding randomisation. |
| Blinding | Blinding was not applicable as this was a retrospective analysis of national data and exposures of interest were not assigned to hospitals by investigators. |

# Reporting for specific materials, systems and methods

We require information from authors about some types of materials, experimental systems and methods used in many studies. Here, indicate whether each material, system or method listed is relevant to your study. If you are not sure if a list item applies to your research, read the appropriate section before selecting a response.

## Materials & experimental systems

| n/a | Involved in the study |
|---|---|
| ☒ | ☐ Antibodies |
| ☒ | ☐ Eukaryotic cell lines |
| ☒ | ☐ Palaeontology and archaeology |
| ☒ | ☐ Animals and other organisms |
| ☐ | ☒ Clinical data |
| ☒ | ☐ Dual use research of concern |

## Methods

| n/a | Involved in the study |
|---|---|
| ☒ | ☐ ChIP-seq |
| ☒ | ☐ Flow cytometry |
| ☒ | ☐ MRI-based neuroimaging |

## Clinical data

Policy information about clinical studies

All manuscripts should comply with the ICMJE guidelines for publication of clinical research and a completed CONSORT checklist must be included with all submissions.

| | |
|---|---|
| Clinical trial registration | Not applicable: this was not a clinical trial. |
| Study protocol | Not applicable: this was retrospective analysis of national data |
| Data collection | Data come from nationally mandated reporting from all NHS acute care hospital trusts in England excluding only those that cared exclusively for children |
| Outcomes | Outcome measures (definite and probable healthcare associated infections) were defined based on ECDC criteria as described in the manuscript. |

