## [Peer Review File · Nature]

Manuscript Title: The burden and dynamics of hospital-acquired SARS-CoV-2 in England

Reviewer Comments & Author Rebuttals

Reviewer Reports on the Initial Version:

Referees' comments:

Referee #1 (Remarks to the Author):

SUMMARY OF KEY RESULTS

This paper combines multiple large datasets in England, together with statistical and mathematical modelling, to make inferences about hospital-acquired SARS-CoV-2 infections in England.

A central argument is that the current definition of hospital-acquired infection is too restrictive and leads to under ascertainment of hospital-acquired cases is convincing. The authors then use data on length-of-stay data to estimate the true number of hospital-acquired infections – this is dependent on many assumptions that I think need further justification. The statistical modelling of hospital-acquired infections is informative and appears robust – the authors have taken measures such as using negative controls for testing their results.

ORIGINALITY

The scale of data – including most trusts in England over an extended period – plus the analysis makes this a valuable contribution to SARS-CoV-2 epidemiology.

CLARITY AND CONTEXT

There are a number of strong statements in the abstract that I think are either not justified by the analysis or need further clarification.

“We show that hospital transmission is likely to have been a major contributor to the burden of COVID-19 in England”

This statement seems to be at odds with a statement in the results “While the level of hospital transmission had little overall impact on an unmitigated epidemic”. The language is unclear (“major contributor” – how big? and ‘burden’ – of infections, deaths, or another measure?). If the sentence in the abstract is correct, could the authors justify it with numbers, e.g. give a % of total cases that are due to hospital transmission.

“nosocomially infected patients likely to have been the main sources of transmission to other patients”

I’m not sure where this is shown in the results.

“reducing hospital transmission could substantially enhance the efficiency of punctuated lockdown

measures in suppressing community transmission”

Could the authors quantify the “substantially enhance” statement.

DATA

This is undoubtedly a rich data source, and there are interesting details which are not described.

For example, there appears to be huge variation in hospital-acquired infections between hospitals, which is mainly used for the statistical model, rather than reported directly. It would be useful for the reader to understand the variation between trusts, with the ranges reported in the results in addition to the national total.

Also, the scales in figure 2 would be useful to understand: e.g. the scale in figure 2a ranges from 0 HAI to 40 HAI per 100 hospital beds. How should that scale/measure be interpreted?

METHODOLOGY

The approach is clearly described. The argument that the current definition misses most cases is convincing.

However, the calculation for the proportion of cases that would be defined as “hospital-acquired” is critical for the conclusions. The authors test assumptions about testing within hospitals, but not testing practices outside hospital, which could lead to different estimates.

- Are length-of-stay and test status independent? Presumably people who test positive in hospital are more likely to have a longer stay? Does this impact the results?

- Is it also assumed that there is an equal chance of testing positive in the community and in hospital? If the probability of testing positive in the community was lower than in hospital (which is plausible because individuals that are released earlier may be healthier – and there is less asymptomatic testing in the community).

SUGGESTED IMPROVEMENTS

- I agree that the current definition is too restrictive. Could the authors suggest how this definition should be improved?

- Although short length of stay leads to a low percentage of cases being identified as “hospital-acquired”, shorter stays are good for breaking the transmission cycle as most infections will be discharged during their incubation period. Did the authors investigate this trade off in their simulations? Could the authors quantify what impact shortening hospital stays would have on transmission in hospitals?

- Could the authors state explicitly what the benefit of testing for all patients at 7-day intervals is?

Other comments:

- Figure 1g: what is the horizontal axis with week number ranging from 25 to 60?
- The maps in figure 2 might want to be presented in a different way as it is easy to identify individual Cornwall and Devon trusts.
- The link for the code is not working https://github.com/BenSCooper/nosocomial_COVID_England

Referee #2 (Remarks to the Author):

See attachment.

Referee #3 (Remarks to the Author):

Cooper et al., review

‘The burden and dynamics of hospital-acquired SARS-CoV-2 in England’

Thank you for the opportunity to review this manuscript by Cooper et al.

Manuscript summary

The authors aim to understand the burden, causes and consequences of SARS-CoV-2 transmission within NHS hospitals in England from June 2020 to February 2021. Importantly, they make causal claims on the drivers of SARS-CoV-2 hospital transmission, to inform how infection prevention and control (IPC) practices can be optimised to reduce hospital spread in future in an evidence-based manner.

Using standard definitions of hospital-acquired COVID-19, they use a dataset of 16,950 and 19,355 probable and definite hospital-acquired infections from NHS Trusts across England, reported via national surveillance and reporting systems. They combine these with other datasets, including healthcare worker (HCW) vaccination status via the SIREN study (<https://snapsurvey.phe.org.uk/siren/>), regional Alpha variant prevalence, and hospital building/architectural information.

In my view, the most important findings from the study are:

- 1-2% of all hospital admissions during this period were likely to have resulted in hospital-acquired COVID-19 infections.
- Based on time-series analysis, patients were thought to mainly be infected with SARS-CoV-2 by other patients; HCW were infected by a mixture of patients and other HCW (consistent with other published studies).
- Factors associated with increased healthcare-associated infections included:
 - o Healthcare-associated infections reported in the preceding week
 - o Reduced hospital side room availability

- o Reduced heated volume per bed (used as a proxy for ventilation)*
- o Newer hospitals (ie. older hospitals were associated with reduced transmission – a surprising finding)
- o Increasing Alpha variant prevalence (in a pre-Delta/ pre-Omicron context)
- o HCW vaccination was associated with a substantial drop in transmission*

*Caveat: similar trend observed for both community and hospital acquired infections

- Hospital transmission can have a substantial impact on sustaining SARS-CoV-2 through community lockdowns.

By determining the factors associated with increased healthcare-associated infections, the authors then suggest optimal strategies for reducing hospital transmission, including:

- Early identification and control of patients with hospital acquired infections
- Reducing asymptomatic transmission, including with improved ventilation, face coverings, increasing patient spacing on wards and limiting bed movements
- Prioritising HCW for vaccination
- Increasing research into reducing airborne transmission eg. via ward design and air filtration systems

Overall impression

This study asks important questions using a comprehensive dataset and there is no doubt it should be published and would be of interest to people working on healthcare infections. It would also be of broader interest to those working in medicine and public health; though the figures are so dense that the central messaging can get lost, reducing the broader appeal to some extent. Editorially, I feel it is probably better suited to a journal more typically associated with clinical medicine - eg. within the Nature portfolio, Nature Medicine seems most appropriate. However, this is an editorial decision and not a comment on scientific quality.

Major comments:

Ethics. There is no Ethics section to the Methods. Which ethical bodies approved the study? What is the information governance framework around accessing and using the data?

Data production and data sharing. I would expect the Methods to begin with data access or data generation - how were all these data produced and accessed? Data generators should be named and acknowledged explicitly. The data availability section states, 'for fields related to numbers of healthcare-associated infections and length of stay, by submitting a Freedom of Information request to NHS England' – does this mean to reproduce the study analysis a scientist would have to submit a FOI request to NHS England? Is there no framework under which the data could be shared with scientists eg under a data sharing agreement? Otherwise the work is not reproducible.

Code availability and reproducibility. The Methods states: 'All analysis code for the current paper is available from https://github.com/BenSCooper/nosocomial_COVID_England.'

I cannot access this link, so cannot review the code or check for reproducibility. This limits the scope of the review.

Clarity of messaging in the figures. The figures are complex and multi-panel to the point of compromising clarity of messaging. The main figures should clearly communicate the key messages from the study. Figure 4 gets to panel u; 21 panels for one figure is too many. I was not clear what

panels j-u were telling me. Figure 3 has 20 separate line plots in panel c, which will be very small on an A4 page. They are unlabelled in the figure; the legend states they are from the 20 largest NHS Trusts, but it's not clear what message the reader is expected to take from these unlabelled plots. The within-plot text for several figures (eg. 3a and legends for 4j-u) would also be very small and hard to read. What is the take-home message being conveyed for each main figure, and is there a more direct and summary way to represent this? Extra content (such as the individual Trust-level plots depicted in figure 3c) can go to supplementary materials.

Minor comments:

Introduction page 5 lines 2-8: It may be helpful to include community-onset healthcare-associated infections in these opening definitions (prior healthcare contact within the 14 days preceding a positive test in the community) - this was often the most challenging category for acute trusts under covid pressure to collect the necessary data on. I don't think this is mentioned anywhere in the paper, though we had significant healthcare-associated community-onset outbreaks (eg. in community dialysis units, and among recently discharged patients re-admitted with Covid).

Introduction page 5 lines 2-3: "understand its dynamical consequences." This could be clarified. Before reading the rest of the paper, it is not clear what 'dynamical consequences' means. Perhaps it could be rephrased to more plainly state the authors' aims for the introductory paragraphs.

I do not believe the authors state why they selected 10th June 2020 - 17th February 2021 for the study. This would be worth including, in introduction or methods.

Can the authors comment on why newer hospitals were associated with more hospital-acquired COVID-19? I would not have predicted this and find it very surprising. In my experience, older hospitals tend to have worse ventilation and more crowded wards, with fewer side rooms. Could this signal have been caused by confounders?

Can the authors expand on what 'heated volume per bed' is and how this relates to ventilation? I think demonstrating that a lack of side rooms and poor ventilation drives healthcare associated infections, ultimately causing patient harm and exacerbating the pandemic, is a very important outcome of this study. It would be good to highlight this more in the discussion.

The authors use a probability distribution for symptom onset by days since infection, with the most probable symptom onset day being 5-6 days post infection (Figure 1a). It would be interesting to add to the discussion how infection parameters such as this may differ between variants - in this study I suspect the distribution for Wuhan and Alpha variants will differ; I suspect Delta and Omicron would differ even more from Wuhan than Alpha. How might these changes affect the results - ie. how generalisable are the IPC recommendations across subsequent/ current/ potential future SARS-CoV-2 variants? The effect of Alpha was so strong (eg. figure 3a), what do the authors think of transmission with Delta and Omicron? Would the authors expect similar improvements to rates of hospital-acquired infections for other respiratory viruses such as influenza, if their recommendations were followed?

Re: the section using community-onset COVID-19 infections as a negative control, to check for

confounding with the hospital-onset associations (page 10, lines 4-9). In my experience, the incidence of hospital-onset SARS-CoV-2 was strongly related to the incidence in the community, for multiple reasons including exhausting side room capacity limiting our ability to isolate patients, and presumably simply more opportunities for within-hospital transmission if there is more SARS-CoV-2 virus around, eg. via HCW infections picked up in the community being introduced to the hospital. Can the authors explain how controlling for community prevalence wouldn't eliminate legitimate signal, given the expected correlation between community prevalence and rates of hospital acquired infections? (In particular for the caveat given on heated volume per bed and HCW vaccination associations with healthcare-associated transmission).

I generally expect a scientific paper to include discussion of the study limitations in the Discussion, and how these may impact on the conclusions/ were addressed.

Some references that could be included in the relevant text:

- Air filtration and airborne pathogen transmission (including SARS-CoV-2) in hospitals: Conway Morris, A., et al. 2021. *Clinical Infectious Diseases*, ciab933, <https://doi.org/10.1093/cid/ciab933>
- Vaccination reducing HCW infections for SARS-CoV-2: Ferris M, et al. 2021. *Elife*. 2021 Nov 16;10:e71131. doi: 10.7554/eLife.71131

Referee #4 (Remarks to the Author):

The COVID-19 pandemic has impacted healthcare facilities worldwide, causing large outbreaks with major morbidity and mortality. However, the actual number of hospital-acquired cases has rarely been quantified at a large scale, notably due to imperfect testing within healthcare settings, especially over the first epidemic waves. In this context, this paper proposes an assessment of the full burden of hospital-acquired SARS-CoV-2 in England between June 2020 and March 2021, identifies the main factors associated with nosocomial SARS-CoV-2 transmission to patients and healthcare workers, and studies the role played by this transmission on the efficiency of lockdown measures in the community.

The authors should be commended for their thorough exploration of the burden of hospital-acquired SARS-CoV-2 in England, combining large-scale data analysis, statistical modelling and dynamic mathematical modelling. This work clearly has important public health implications and should prove key in justifying further studies to help control SARS-CoV-2 transmission within healthcare settings.

There are however several points that I think should be addressed to benefit the paper.

Regarding the burden quantification

Although detailed, the description of the method to quantify the number of hospital-acquired SARS-CoV-2 infections is at times unclear and hard to follow. This is, I believe, due to several reasons:

- The use, at times, of different terms for the same concept (e.g., “distribution of times from infection to onset of symptoms” vs. “incubation period distribution”);
- A lack of explicit equations in some parts of the section to help the reader understand the approach, rather than general sentences e.g. “In general, P_{ij} will depend on...”;

- Several repetitions, e.g. the mention that the definition of “definite” HAIs requires the first positive sample to be taken 15 days or more after hospital admission, which is made at least twice in the same section; or the definition of γ_{ijmd} which is provided twice, using slightly different terms, on the same page;
- Some confusing notations, e.g. on lines 6 and 8 of p. 23 π_{ij} is used to denote both the probability that a new nosocomial infection in trust i and week j is detected and the probability that a new nosocomial infection in trust i and week j is detected and meets the HAI definition; maybe the latter should read π'_{ij} ?

Overall, I would recommend reorganizing this method section to start from the most general equation (how the total number of HAIs z_{ij} is computed from the observed number of HAIs y_{ij} and the probability π_{ij} that an occurring infection in trust i and week j will be detected), and then to go on to explain how π_{ij} is computed from γ_{ijmd} and P_{ijmd} (again providing equations straight away), and then only explain how these two components are computed. Adding subtitles to clearly delineate each component’s calculation could also maybe help.

In addition, while efforts were clearly made to explore some uncertainties in this quantification through parameter distributions and sensitivity analyses, this is not the case for the assumed PCR test sensitivity over time since infection, which is informed by the paper from Hellewell et al. Indeed, this study is based on data on self-performed tests in healthcare workers, who are likely to have less severe forms of COVID-19 than patients, for whom the sensitivity curve is used here. How would using PCR sensitivity over time data from other papers (e.g., the study by Kucirka et al.), which are based on healthcare-worker-performed tests on patients, impact the results? I believe this would lead to lower probabilities of detection before day 3, but higher probabilities of detection after day 7, with potential consequences on the overall estimates.

Regarding the statistical analysis

The authors underline that “both heated volume per bed and HCW vaccination coverage show similar negative associations with the control outcome as for reported healthcare-associated infection outcomes”. However, no possible explanation is proposed for this finding. Could this be related to hospital type for instance, with higher vaccination coverage and more admissions of COVID-19 patients in large university hospitals? It could be interesting to include a paragraph discussing possible interpretations in the discussion.

Regarding the mathematical modelling

A general description of the model is lacking from the methods, which only includes a description of the scenarios explored, followed straight away by the model equations. Why include these equations in the Methods but put the model diagram and lists of model variables and parameters in the Supplement? I’m afraid that these equations are impossible to understand without these lists. I would suggest putting the model diagram in the Methods, along with a paragraph briefly describing the main features of the model (main populations involved and main disease stages considered, who

is assumed to be in contact with whom etc.), but moving the equations to the Supplement.
As a minor remark, the equation on SH is written twice on p.31.

Author Rebuttals to Initial Comments:

Response to referees' comments

We would like to thank all the referees for their encouraging and constructive comments and for taking time to read the manuscript with such care. In our responses below we reproduce the referee comments we are responding to in blue italics, and our responses to them are in black.

Referees' comments:

Referee #1 (Remarks to the Author):

SUMMARY OF KEY RESULTS

This paper combines multiple large datasets in England, together with statistical and mathematical modelling, to make inferences about hospital-acquired SARS-CoV-2 infections in England.

A central argument is that the current definition of hospital-acquired infection is too restrictive and leads to under ascertainment of hospital-acquired cases is convincing. The authors then use data on length-of-stay data to estimate the true number of hospital-acquired infections – this is dependent on many assumptions that I think need further justification. The statistical modelling of hospital-acquired infections is informative and appears robust – the authors have taken measures such as using negative controls for testing their results.

ORIGINALITY

The scale of data – including most trusts in England over an extended period – plus the analysis makes this a valuable contribution to SARS-CoV-2 epidemiology.

Response:

We thank the reviewer for the positive comments. Further justification of assumptions has now been added, and is described in the more specific comments below.

CLARITY AND CONTEXT

There are a number of strong statements in the abstract that I think are either not justified by the analysis or need further clarification.

“We show that hospital transmission is likely to have been a major contributor to the burden of COVID-19 in England”

This statement seems to be at odds with a statement in the results “While the level of hospital transmission had little overall impact on an unmitigated epidemic”. The language is unclear (“major contributor” – how big? and ‘burden’ – of infections, deaths, or another measure?). If the sentence in the abstract is correct, could the authors justify it with numbers, e.g. give a % of total cases that are due to hospital transmission.

Response:

We agree that there was a lack of precision in our use of language here. We meant “deaths” in light of reports of case fatality rates of 20-30% in hospital outbreaks, even amongst vaccinated populations and the multiple risk factors amongst hospitalised patients which would be expected to translate to high case fatality [1,2]. However, we recognise that quantifying attributable mortality from infections in patients who are already hospitalised is challenging (and is the subject of ongoing work) so rather than speculate about the likely magnitude of attributable mortality amongst nosocomially infected patients we have deleted this sentence.

[1] Shitrit Prina, Zuckerman Neta S, Mor Orna, Gottesman Bat-Sheva, Chowers Michal. Nosocomial outbreak caused by the SARS- CoV-2 Delta variant in a highly vaccinated population, Israel, July 2021. Euro Surveill. 2021;26(39):pii=2100822. [https://doi.org/ 10.2807/1560-7917.ES.2021.26.39.2100822](https://doi.org/10.2807/1560-7917.ES.2021.26.39.2100822)

[2] Hetemäki Iivo, Kääriäinen Sohvi, Alho Pirjo, Mikkola Janne, Savolainen-Kopra Carita, Ikonen Niina, Nohynek Hanna, Lyytikäinen Outi. An outbreak caused by the SARS-CoV-2 Delta variant (B.1.617.2) in a secondary care hospital in Finland, May 2021. Euro Surveill. 2021;26(30):pii=2100636. <https://doi.org/10.2807/1560-7917.ES.2021.26.30.2100636>

“nosocomially infected patients likely to have been the main sources of transmission to other patients”

I’m not sure where this is shown in the results.

Response:

This is from Figure 3f. We have rephrased this section to avoid the vague term “likely” and instead emphasise that this analysis contributes to the totality of evidence. We also expand on this in the discussion [lines 366-376] , highlighting how high resolution data also provides support for this..

“reducing hospital transmission could substantially enhance the efficiency of punctuated lockdown measures in suppressing community transmission”

Could the authors quantify the “substantially enhance” statement.

Response:

The potential magnitude of the effect under a variety of scenarios (varying levels of hospital transmission and thresholds for imposing and releasing lockdowns) are explored in Figure 4 and in the accompanying text (page 15), showing that in some cases when epidemics are controlled through punctuated lockdowns a 50% reduction in hospital transmission can reduce the community attack rate from 27% to 12%. We haven’t been able to find a way to quantify this statement within the abstract within the word limit (without substantial changes to other parts of the abstract), but think that a reduction from 27% to 12% could reasonably be described as substantial.

DATA

This is undoubtedly a rich data source, and there are interesting details which are not described. For example, there appears to be huge variation in hospital-acquired

infections between hospitals, which is mainly used for the statistical model, rather than reported directly. It would be useful for the reader to understand the variation between trusts, with the ranges reported in the results in addition to the national total.

Response:

We agree. We have now added this information as Extended data Figure 1 showing this variation both in cumulative SARS-CoV-2 infections over the study period and in trust-level covariates used in the regression analysis and correlations between these variables using a pairs plot. We have also added text to the results section of the main manuscript that summarises this information and refers to Extended data Figure 1 (lines 157-166).

Also, the scales in figure 2 would be useful to understand: e.g. the scale in figure 2a ranges from 0 HAI to 40 HAI per 100 hospital beds. How should that scale/measure be interpreted?

Response:

Rates are expressed as per 100 hospital beds as data on patient days at risk of infection were not available. Expressed this way also allows easy contrast with community-acquired SARS-CoV-2 hospital admissions which we also express as a number per 100 beds (without this adjustment much of the variation would simply reflect variation in hospital trust size as larger trusts would be expected to admit more SARS-CoV-2 infected patients). Median (IQR) bed occupancy during the study period was 83% (78%, 88%) and the data on HAI correspond to a median (IQR) rate of 0.8 (0.5, 1.1) “definite” hospital associated infections per 1000 occupied bed days and 0.9 (0.5, 1.3) “probable” hospital associated infections per 1000 occupied bed days. We have now added to the first sentence of the results (page 6) so that it now provides this information on the number of hospital associated infections per occupied bed day.

METHODOLOGY

The approach is clearly described. The argument that the current definition misses most cases is convincing. However, the calculation for the proportion of cases that would be defined as “hospital-acquired” is critical for the conclusions. The authors test assumptions about testing within hospitals, but not testing practices outside hospital, which could lead to different estimates.

Response:

The primary analysis estimated total numbers of hospital-acquired infections using data on “definite healthcare-associated” infections (those with a delay of 15 or more days between hospital admission and PCR-confirmed infection) because the number of such “definite” nosocomial infections will be minimally affected by testing

in the community. This is because incubation periods longer than 14 days are extremely rare (for example, Lauer et al. estimated that only 1 in 100 cases have an incubation period longer than 14 days <https://www.acpjournals.org/doi/full/10.7326/M20-0504>), and even for the small number of community-acquired cases with an incubation period of over 14 days, to first test positive 15 or more days after admission would require at least three false negative PCR tests (on admission and at day 3 and 6 under the standard screening protocol) and would also require hospital admission (for a reason other than covid) shortly after infection. We therefore do not expect these estimates (based on definite hospital associated infections) to be affected in a meaningful way by testing in the community and assumptions about community testing do not come into the analysis. One of the sensitivity analyses we performed made use of probable and definite hospital associated infections (adding those infections which were first PCR positive 8-14 days after hospital admission and classified as “probable healthcare associated”). This gave similar estimates to the main analysis, though overall estimates were 20-30% higher. Genomic analysis indicates that the great majority of infections classified as “probable hospital-associated” are also acquired in hospital (eg. see Stirrup et al <https://doi.org/10.1101/2022.02.10.22270799>), but since these “probable hospital associated” cases are also compatible with community onset cases with moderate incubation periods in this case we would expect assumptions about community testing of asymptomatic individuals to have some impact – the more community testing of asymptomatic patients (for example, in care homes) there is then the higher the proportion of these “probable hospital-associated” cases we would expect to result from hospital transmission. However, given that this is only a sensitivity analysis and we don’t have any reliable information sources on testing of asymptomatic individuals representative of the population admitted to hospital for non-covid reasons we think there would be little value in further analysis in this case. If advised to by editors (and given the space to do so) we could certainly expand this discussion to address this point, though because the main analysis is based on a 15-day cutoff we do not consider this to be an important limitation of the primary analysis.

Are length-of-stay and test status independent? Presumably people who test positive in hospital are more likely to have a longer stay? Does this impact the results?

Response:

Dependencies between length of stay and test status can arise in a number of ways. First of all, hospital-acquired covid infection could cause an increased length of stay for clinical reasons, and a longer length of stay will increase the chance of being tested. Secondly, a positive test could increase length of stay for non-clinical reasons (e.g. difficulty discharging a PCR+ve patient). Thirdly, there could be confounding factors causally linked to both length of stay and testing for reasons other than SARS-CoV-2 (for example, presence of a respiratory infection caused by a different pathogen that might both prolong stay and give rise to symptoms that prompt PCR testing). We do not think any of these dependencies would impact

results in this paper as we make the conservative assumption that all patients who become symptomatic while in hospital are tested. Had we made the less conservative assumption (less conservative in the sense that it would give rise to higher estimates of hospital-acquired infection) that some symptomatic hospitalised patients were not tested because they were discharged before the chance to test arose, then additional length of stay caused by SARS-CoV-2 would have had some small impact. In addition, these dependencies would be important to consider when trying to estimate the impact of a hospital-acquired SARS-CoV-2 infection on length of stay. We don't, however, address this question in this paper, though a manuscript is in preparation where we apply an approach using inverse probability weights to account for the potential informative censoring introduced by treating SARS-CoV-2 infections as censoring events, and estimate an increased length of stay of 1-2 days caused by nosocomial SARS-CoV-2 infection.

- Is it also assumed that there is an equal chance of testing positive in the community and in hospital? If the probability of testing positive in the community was lower than in hospital (which is plausible because individuals that are released earlier may be healthier – and there is less asymptomatic testing in the community).

Response:

As mentioned above, our analysis makes no assumptions about the chance of testing positive in the community, which was known to change substantially over time.

SUGGESTED IMPROVEMENTS

- I agree that the current definition is too restrictive. Could the authors suggest how this definition should be improved?

Response:

Our feeling is that as long as it is recognised that those infections meeting the definitions of “definite” or “probable” hospital-associated infection will inevitably exclude many other hospital-acquired infections the current classification into “definite”, “probable” and “indeterminate” is sensible.

- Although short length of stay leads to a low percentage of cases being identified as “hospital-acquired”, shorter stays are good for breaking the transmission cycle as most infections will be discharged during their incubation period. Did the authors investigate this trade off in their simulations? Could the authors quantify what impact shortening hospital stays would have on transmission in hospitals?

Response:

We agree that there are trade-offs to consider when considering length of stay, but the epidemiological picture is complex as it would also be important to consider the implications for the locations patients are discharged to (which may include care homes, or homes with other vulnerable people). Given this complexity, we consider

this an interesting question to be addressed in future work, but beyond the scope of the current paper.

- Could the authors state explicitly what the benefit of testing for all patients at 7-day intervals is?

Response:

We don't explicitly address the pros and cons of different testing policies in this paper, though we have considered such questions in other research to come out of our team (see Evans et al *Phil Trans Roy Soc B* <https://doi.org/10.1098/rstb.2020.0268>). However, Figure 1D does show how the probability of detecting hospital acquired infections changes with the addition of 7-day testing, giving approximately a 10% absolute increase in the chance of detecting a hospital-acquired infection.

Other comments:

• *Figure 1g: what is the horizontal axis with week number ranging from 25 to 60?*

Response:

This was previously explained only in the methods "counting week numbers as one plus the number of complete seven day periods since January 1st 2020" but not in the figure caption. We have now amended the caption so the information is also given there.

The maps in figure 2 might want to be presented in a different way as it is easy to identify individual Cornwall and Devon trusts.

Since the trust-level data used here are already in the public domain, we can confirm that this is not a concern.

• *The link for the code is not working https://github.com/BenSCooper/nosocomial_COVID_England*

Response:

This has now been fixed.

-

Referee # 2 (Remarks to the Author):

This manuscript first used a Bayesian statistical model to estimate weekly numbers of hospital-acquired infections using reported weekly numbers of confirmed hospital-associated cases based on ECDC criteria, together with distributions of length-of-stay, incubation period, and test sensitivity. The authors then imputed weekly numbers of HCW infections from the time series of COVID-19-related

absence data of HCW. Using these imputed case numbers, an autoregressive GLM was used to identify driving factors and quantify contributing infectious sources for the risk of hospital-acquired infections among patients and HCWs. Finally, the authors used a deterministic model to assess the role of reducing hospital transmission in the overall transmission within and between hospitals and the community at large, with or without lockdown in the community. Overall, this manuscript is well written, and the results are interesting, offering some insights about what drove hospital-associated transmissions to patients and HCWs, and under what conditions hospital transmission control can be influential to the overall epidemic in the society. Several issues, however, need clarification to better understand the validity of the model. I recommend major revision.

Response:

We thank the reviewer for the positive comments. We have sought to clarify the highlighted issues below.

Do the GLMs take into account uncertainty in the estimation of weekly counts of hospital-acquired infections among patients and HCWs? If not, the statistical significance of the covariate effects in the GLMs would be questionable.

Response:

We initially attempted to do this, representing the underlying infection dynamics as a latent process with observed infections stochastically depending on the latent state (i.e a type of hidden Markov model). However, we had to abandon this approach as we were unable to obtain reliable inferences; we do, however, agree that this would be an interesting (though challenging) area for future work. Instead we adopted an alternative approach to guide our interpretation of regression coefficients in fitted models: we generated synthetic data from a latent process model using specified negative binomial offspring distributions, and sampled the generation interval from a Weibull distribution with shape 2.83 and scale 5.67 (taken from Ferretti, L. et al. 2020). We then sampled the number of observed infections from the synthetic latent process data using binomial distributions where the “probability of success” was the assumed probability of observing an infection in a given host category (patient or healthcare worker). We did this over a grid of values for reproduction numbers and observation probabilities, then applied our GLM to the synthetic data and compared regression coefficients with parameter values used to generate the synthetic data. The procedure and results are reported in section 2.3 “Generation and analysis of synthetic data” of the Supplementary Information and Extended data Figure 4, and also referred to in the results section (page 11) of the main manuscript in the paragraph beginning “To help interpret estimated regression coefficients we performed a series of simulation studies” . We note that in the original submission we failed to correctly refer to the detailed methods in this part of the text. We have now corrected this, adding text referring to “(Supplementary Information: section 2.3)”. The results show

monotonic relationships between the probabilities of observing infections and estimated regression coefficients, and also highlight the problem the reviewer alludes to – i.e. regression coefficients cannot simply be interpreted as reproduction numbers. Note that we don't refer to "Statistical significance" anywhere in the manuscript for reasons outlined in the ASA's consensus statement on p-values <https://www.tandfonline.com/doi/full/10.1080/00031305.2016.1154108>

In the GLMs, the trust-specific intercept is time-invariant, and time-dependent nonpharmaceutical interventions such as lockdowns are not accounted for.

Response:

Time-dependent NPIs such as lockdowns are accounted for in the model by time-dependent variation in the (1) admission of patients infected in the community and (2) number of infected HCWs. Community-level nonpharmaceutical interventions such as lockdowns would not otherwise be expected to directly affect the risk of patients becoming infected in the hospital. Also, hospital infection control guidelines (including testing and use of PPE) did not change over the study period. We do, however, agree that there may have been other factors (not directly linked to national guidance) that may have changed over the study period in a way that is not captured by the SARS-CoV-2 admissions variable. We have therefore carried out additional sensitivity analysis using a model with an additional spline component to account for time-varying changes in the expected number of infections that are not accounted for by model covariates. We refer to this model as P1.1.1.tv and define it on page 31 of the main manuscript: Results are shown in Table S18 in the SI and the estimated spline functions are given in Extended data Figure 6. Note that the leave-one-out information criterion strongly favours the model without the spline component (P1.1.1) which has a leave-one-out information criterion of 8884.7 versus 8968.89 for the model with the spline component. While the two models give broadly similar results (showing negative associations between the number of hospital-acquired infections and numbers of side rooms and heated volume per bed) the strong negative association between vaccine coverage and number of nosocomial infections seen in the simpler model is not seen in the spline model (with the spline function instead showing a downward trend at about the time the vaccine was introduced (Extended data Figure 6)). While other model formulations with spline terms are possible (for example, by adding a spline term as an additive rather than multiplicative effect) we were unable to run such models in *Stan* without a large number of diverges indicating that reliable inference was not possible. The most likely reason for this is that the models did not give a good fit to the data.

What was vaccine coverage among patients during the study period? Was it truly negligible?

Response:

The regression analysis is restricted to the period between week 42 (beginning 14th October 2020) and week 55 (beginning 13th January 2021). In England, vaccine

rollout to the over 70s and clinically extremely vulnerable began on 18 January 2021, while residents in care homes for older adults and their carers and all those aged 80 and over were first eligible for vaccination on 8th December 2020.

If we assume that those vaccinated with the first dose two or more weeks previously have some degree of immunity, we can consider anyone vaccinated by the end of December 2020 to be partially vaccine-protected by the start of week 55. NHS England reported the total number of people in England who had been vaccinated by 27th December to be 786,000, and these vaccinations occurred in the following priority groups: care home residents and their carers, frontline health and social care workers and people aged 80 years old and over. 524,439 of those vaccinated by this time point are reported to be aged 80 and over. In 2020 there were 2.9 million people aged 80 years or over living in England, so 18% of those aged 80 and over can be assumed to have had some degree of vaccine protection by week 55. Of the 261,561 people who were vaccinated by 27th December and who were not aged 80 or over, we do not have a breakdown of the number who were care home residents, carers etc but at this time the number of people in England aged 70-79 was 4.8 million so at most 10% (i.e. $786000 / (4800000 + 2900000)$) of those aged 70 and over in England had been vaccinated with at least one dose two or more weeks previously by week 55. Though this coverage is small and protection shortly after a single dose will be quite limited, we think it is important to highlight this as a limitation and have therefore added text to the discussion (lines 389-395) discussing this limitation and have also provided the underlying reasoning and references to data sources in the SI, section 2.4:

4.

To estimate the number of hospital-acquired infections, a key step is the estimation of γ_{ijmd} , the probability that, given a new hospital-acquired infection in trust i in week j occurs, it occurs in a patient with length of stay m on day of stay d . γ_{ijmd} is approximated by ψ_{imd} , the expected proportion of patients who both have a length of stay of m days and are currently on day of stay d on a random given day. How good is this approximation? As ψ_{imd} depends on λ_{im} , the estimation of λ_{im} should then be ideally based on patients who tested negative at admission but tested positive during hospitalization. However, it doesn't seem to be the case. On page 25, it states "Length-of-stay distributions ... excluding: i) patients who were admitted with PCR confirmed COVID-19, ii) patients who had samples taken in the first seven days of their hospital stay which were PCR positive for SARS-CoV-2; and iii) patients with a length-of-stay of less than one day." Why is exclusion criterion (ii) imposed?

Response:

Exclusion criterion ii) was imposed for a similar reason to exclusion criterion i) – these are potentially patients with a community-acquired infection which would mean they are not at risk of hospital-acquired infection. However, since these patients are (not unreasonably) classified as “indeterminate” hospital associated, we agree that there is also a case for including them. We have therefore now added a

sensitivity analysis where these patients are included. This results in very slight changes in the estimated number of nosocomial infections, with the upper bound (90% CrI) decreasing from 143,000 (123,000, 167,000) to 142,000 (123,000, 166,000) and the lower bound decreasing from 99,000 (95,000, 104,000) to 98,000 (94,000, 103,000). We have now modified the main manuscript so that it refers to this additional sensitivity analysis (page 7), and give results of this new sensitivity analysis in section 2.1 of the SI.

Not sure how exactly the negative-binomial GLMs inform the parameters of the deterministic hospital-community transmission model. Did you calibrate the deterministic model to observed data for parameter estimation? The calibration may not be easy, but it should be explained clearly how the parameters of the deterministic model were identified.

Response:

We did not attempt to formally fit the deterministic model to data as this work was conducted “with the aim of providing qualitative insights” (as we state on line 756). That is, the intention of this part of the work was not to reconstruct the dynamics in England but rather to illustrate the potential importance of nosocomial transmission to overall dynamics when transmission in the community is curtailed by punctuated lockdowns. The parameters are therefore intended to be illustrative of plausible ranges consistent with the findings of the analysis from this paper and the wider literature about nosocomial SARS-CoV-2 transmission in the UK. In particular we make use of our analysis of high resolution data from Oxford University Hospitals (Figure 4 from, Mo Yin et al, PloS Med 2021 <https://doi.org/10.1371/journal.pmed.1003816>), and the largest genomic study of nosocomial transmission of SARS-CoV-2 (<https://www.nature.com/articles/s41467-022-28291-y>); all these studies point to substantially higher transmission from nosocomially infected patients compared to patients admitted with COVID-19 from the community. These parameters are also in line with those used in a previously published model that was calibrated using national data on hospital admissions and HCW infection rates (<https://royalsocietypublishing.org/doi/pdf/10.1098/rstb.2020.0268>). To make this clearer we have now modified the main article text to make it clear that the model is informed by the both the estimates in this paper and by the wider literature (lines 303-304).

The deterministic model structure is not explained adequately. For example, the legend of extended data Fig 1 says “two asymptomatic infectious states (I1, I2)”. Is I2 really an asymptomatic state? If so, as I' is severe symptomatic disease, I don't see a moderate symptomatic disease state which is necessary.

Response:

We agree, and have substantially rewritten the legend to the flow diagram and expanded the Methods section (page 34) which now explains the structure of the

model. We have also reorganised this section, including the flow diagram in the methods section (Figure 5) and moved the equations to the SI (as suggested by reviewer 4). It was incorrect of us to refer to I1 and I2 as “asymptomatic infectious states” and this has now been fixed.

Minor comments

1. The line numbers are very confusing as I resets every 10 lines.

Response:

Apologies. This may have been due to a left hand margin that was too small. This has now been fixed.

2. *The different testing policies are not explained clearly. For example, on page 24, “A second testing policy extends this by assuming that in the event of a negative screening result from a patient with symptoms, daily testing will continue to be performed until patient discharge, the first positive test, or three consecutive negative tests (whichever occurs first)”. Does this stopping rule also apply to other policies, such as weekly, twice weekly, and three times weekly?*

Response:

Only the policy labelled “Symptomatic testing, retesting if negative” in figure 1D involves retesting if negative. We have now modified the text (page 28) to make this clear: “We consider additional testing policies which combine symptomatic testing (without retesting if negative)....”

On page 6, 2nd paragraph, two policies are described. The first one, what does “but includes no asymptomatic testing after day seven post-admission” mean exactly? What happens during the first week of admission under this assumption?

Response:

Apologies for lack of clarity. This referred to the policy described in the previous paragraph with “asymptomatic PCR testing on days of stay 3 and 6 (as recommended by national screening guidance in England)”. To make this clearer we have modified the text (page 7) so that it now reads “First, we assume patient testing followed national guidance at the time which specified testing of symptomatic patients (without retesting) and included asymptomatic testing on two occasions in the first week but none after day seven post-admission.

The second one, testing at seven-day intervals post-admission, I assume it means weekly asymptomatic testing. Why does it provide a upper bound for the chance of identifying hospital-acquired infections? Why wouldn't daily testing

be?

Response:

This is because, as Figure 1D shows, it has a higher chance of identifying hospital-acquired infections than the first one (which corresponds to national screening guidance), and to our knowledge no hospitals were testing asymptomatic patients at a higher frequency than this (had daily testing been used, that would of course have given a higher chance of identifying hospital-acquired infections). To make this clearer we have now added a comment on page 7 to say that this was the “maximal testing policy known to be used in practice”.

3. *Fig 1(d), what’s the difference between “symptomatic testing + weekly screening” and “symptomatic testing + day 7, 13, 21 ... screening”?*

Response:

The former is on fixed days of the week, while the latter is at fixed times after hospital admission. This is stated in the caption to Figure 1 “the policy of screening all patients at seven day intervals after admission is highlighted in blue (note that in contrast to this policy, weekly and 2 and 3 x weekly policies screen on fixed days of the week)”

4. *Tables S15-S17 are difficult to interpret.*

Response:

Please see the response to Reviewer 3’s questions about the use of the negative control.

5. *Fig 3 e-g, make it clear that these contributions are evaluated with vaccination of HCWs.*

Response:

This has now been added to the caption to Figure 3 “accounting for HCW vaccination and Alpha variant effects”

6. *Ext. Fig. 5, I believe (a) and (b) are switched. The lower panel (b) should correspond to no mitigation. The upper panel should be discussed in the results section (page 14) explicitly. In this panel, is lockdown implemented throughout without lifting? This is important, as it implies hospital transmission does not affect the overall epidemic much either without lockdown or with complete lockdown, and it does make a difference only*

when lockdown is punctuated.

Response:

Thank you for spotting this! This is now fixed and the caption has been changed to make it clear it is not lifted. The text on page 15 has also been changed to refer to both panels.

7. *Page 14, line 8 in the middle of the page, “with corresponding increases” actually means “with corresponding decreases”?*

Response:

Thank you. This has now been corrected.

8. *Fig 4 s-u, what is the reference setting for calculating the number of averted cases, no lockdown at all in the community for each hospital transmission intensity?*

Response:

Yes, and we now make it clear that the reference is an unmitigated epidemic on lines 345-348.

9. *Page 16, in the middle, “superspreading is implicit ..., which attribute 80% of detected... to 21% of infected patients”, where exactly is this result shown?*

Response:

This was not shown anywhere in the previous version but we now show the proportion of new infections attributed to a given proportion of the most infectious cases based on results for “probable and definite” healthcare associated infections and “definite” healthcare associated infections in Extended data Figure S7. The 21% figure comes from model P1.1.1 and the “probable and definite” outcome, but in all cases close to 20% of infected patients account for 80% of transmission from patients. We have also modified the text in the discussion (page 18) which now reads:

“superspreading is implicit in our negative binomial models which attribute 80% of detected patient-patient transmission events from nosocomially-infected patients to approximately 20% of infected patients (Extended Data Fig. S7).”

10. *Please comment on the role of outpatients in hospital transmissions in the discussion.*

Response:

We have added this sentence to the discussion (lines 395-397) “We did not consider outpatients in this work as they are typically cared for in separate outpatient clinic settings distinct from the wards of acute hospitals.”

11. Page 23, I'm wondering why $\psi_{imd} = \lambda_{im}m / (d \sum_n \lambda_{in}n)$. Shouldn't it be $\psi_{imd} = \left[\frac{\lambda_{im}m}{\sum_n \lambda_{in}n} \right] \frac{1}{m} I(m \geq d)$? Here $I(m \geq d)$ is an indicator function. $\frac{\lambda_{im}m}{\sum_n \lambda_{in}n}$ is the probability that on a randomly chosen day a randomly chosen patient has a length of stay m days, and $\frac{1}{m}$ is the probability that this randomly chosen day is day d of stay. Am I missing something?

Response:

Thank you for spotting this. Indeed, the d in the denominator should have been an m . We have checked that the code is not affected by this, corrected the typo in the methods, and rewritten the expression using the indicator function as suggested. The corrected expression now appears on page 27 as there has been some re-organisation of this section in response to a suggestion from reviewer 4.

- 11. Page 27, about the rationale for the form of dispersion term, as some hospitals may not be very large, especially when the number of infections in the previous week is not trivial, the argument for is not very convincing. You may simply compare the current model to the model with a constant*

Response:

We did initially attempt to fit models with a constant dispersion term but were unable to fit these models without a large number divergent transitions after warmup, indicating that reliable inference was not possible with these models.

- 12. Page 29, in the middle, I'd suggest writing a formula explicitly, e.g., $a_t - x_{t-10} + x_t = a_{t+1}$, where a_t is the number absent HCWs and x_t is the number entering isolation on day t .*

Response:

We have made this change as suggested (now on page 33).

Referee #3 (Remarks to the Author):

Cooper et al., review

'The burden and dynamics of hospital-acquired SARS-CoV-2 in England'

*Thank you for the opportunity to review this manuscript by Cooper et al.
Manuscript summary*

The authors aim to understand the burden, causes and consequences of SARS-CoV-2 transmission within NHS hospitals in England from June 2020 to February 2021. Importantly, they make causal claims on the drivers of SARS-CoV-2 hospital transmission, to inform how infection prevention and control (IPC) practices can be optimised to reduce hospital spread in future in an evidence-based manner.

Using standard definitions of hospital-acquired COVID-19, they use a dataset of 16,950 and 19,355 probable and definite hospital-acquired infections from NHS Trusts across England, reported via national surveillance and reporting systems. They combine these with other datasets, including healthcare worker (HCW) vaccination status via the SIREN study (<https://snapsurvey.phe.org.uk/siren/>), regional Alpha variant prevalence, and hospital building/ architectural information.

In my view, the most important findings from the study are:

- 1-2% of all hospital admissions during this period were likely to have resulted in hospital-acquired COVID-19 infections.*
 - Based on time-series analysis, patients were thought to mainly be infected with SARS-CoV-2 by other patients; HCW were infected by a mixture of patients and other HCW (consistent with other published studies).*
 - Factors associated with increased healthcare-associated infections included:*
 - o Healthcare-associated infections reported in the preceding week*
 - o Reduced hospital side room availability*
 - o Reduced heated volume per bed (used as a proxy for ventilation)**
 - o Newer hospitals (ie. older hospitals were associated with reduced transmission – a surprising finding)*
 - o Increasing Alpha variant prevalence (in a pre-Delta/ pre-Omicron context)*
 - o HCW vaccination was associated with a substantial drop in transmission**
- *Caveat: similar trend observed for both community and hospital acquired infections*
- Hospital transmission can have a substantial impact on sustaining SARS-CoV-2 through community lockdowns.*

By determining the factors associated with increased healthcare-associated infections, the authors then suggest optimal strategies for reducing hospital transmission, including:

- Early identification and control of patients with hospital acquired infections*
- Reducing asymptomatic transmission, including with improved ventilation, face coverings, increasing patient spacing on wards and limiting bed movements*
- Prioritising HCW for vaccination*
- Increasing research into reducing airborne transmission eg. via ward design and air filtration systems*

Overall impression

This study asks important questions using a comprehensive dataset and there is no doubt it should be published and would be of interest to people working on healthcare infections. It would also be of broader interest to those working in medicine and public health; though the figures are so dense that the central messaging can get lost, reducing the broader appeal to some extent. Editorially, I feel it is probably better suited to a journal more typically associated with clinical

medicine - eg. within the Nature portfolio, Nature Medicine seems most appropriate. However, this is an editorial decision and not a comment on scientific quality.

Response:

We thank the reviewer for the positive comments.

Major comments:

Ethics. There is no Ethics section to the Methods. Which ethical bodies approved the study? What is the information governance framework around accessing and using the data?

Response:

We have now added an Ethics section at the start of the Methods.

Data production and data sharing. I would expect the Methods to begin with data access or data generation - how were all these data produced and accessed? Data generators should be named and acknowledged explicitly. The data availability section states, 'for fields related to numbers of healthcare-associated infections and length of stay, by submitting a Freedom of Information request to NHS England' – does this mean to reproduce the study analysis a scientist would have to submit a FOI request to NHS England? Is there no framework under which the data could be shared with scientists eg under a data sharing agreement? Otherwise the work is not reproducible.

Response:

Apologies if we have misunderstood formatting requirements, but our understanding was that the data access information should go in the “data availability statement” which can be found on pages 35-37. We would of course be happy to reproduce this text in the Methods section if it was felt appropriate. Regarding data availability, we have made extensive efforts over several weeks to obtain permission from NHS England to allow us to share all the data used in the analysis, but — frustratingly — we have just heard that they are not going to grant us this permission. While the data can in principle be obtained via a FOI request (as mentioned in the original submission – and, indeed, journalists have successfully obtained such infection data via this route) this may not be the optimal route to obtaining the data and we have therefore modified the data availability statement so that it now reads:

“Infection data used for this analysis were taken from daily situation reports between 10th June 2020 and 17th February 2021 and shared privately with the Scientific Pandemic Influenza Group on Modelling (SPI-M).....

COVID-19 admission data for NHS trusts are publicly available by direct download from <https://www.england.nhs.uk/statistics/statistical-work-areas/covid-19-hospital-activity/>. We do not have permission to share data on healthcare-associated infections and length of stay distributions, and requests for these should be sent to NHS England.”

To ensure transparency we have therefore created a synthetic data set which replaces fields which are currently only available via a formal request to NHS England with synthetic data obtained by simulating data from a saturated Poisson model. The synthetic data and the code used to generate them are available in the github repository (synthetic_sitreps_eng_expanded.rds, synthetic_vacc_cov_by_region.csv, Create synthetic data.R).

Code availability and reproducibility. The Methods states: 'All analysis code for the current paper is available from https://github.com/BenSCooper/nosocomial_COVID_England.' I cannot access this link, so cannot review the code or check for reproducibility. This limits the scope of the review.

Response:

Apologies. This has now been fixed.

Clarity of messaging in the figures. The figures are complex and multi-panel to the point of compromising clarity of messaging. The main figures should clearly communicate the key messages from the study. Figure 4 gets to panel u; 21 panels for one figure is too many. I was not clear what panels j-u were telling me. Figure 3 has 20 separate line plots in panel c, which will be very small on an A4 page. They are unlabelled in the figure; the legend states they are from the 20 largest NHS Trusts, but it's not clear what message the reader is expected to take from these unlabelled plots. The within-plot text for several figures (eg. 3a and legends for 4j-u) would also be very small and hard to read. What is the take-home message being conveyed for each main figure, and is there a more direct and summary way to represent this? Extra content (such as the individual Trust-level plots depicted in figure 3c) can go to supplementary materials.

Response:

We thank the reviewer for these constructive comments. Regarding Figure 3 our personal preference is always to show something as close to the original data as possible together with model fits. While we felt this was not feasible to do this for all the NHS trusts in the main paper due to space limitations, we felt showing the largest 20 and the rest in the supplementary material would be a good compromise. We would of course be open to simplifying this if the paper is accepted and if journal editors felt this was necessary. We would also be open to moving parts j-u of figure 4 to the SI with a lengthier explanation if the editors feel it is appropriate. Clearly there is a subjective element to these decisions, and if the paper is accepted for publication we will be keen to be guided by editors in the formatting of figures (and deciding what gets relegated to the SI) taking on board the above comments as well as those of the other reviewers.

Minor comments:

Introduction page 5 lines 2-8: It may be helpful to include community-onset healthcare-associated infections in these opening definitions (prior healthcare

contact within the 14 days preceding a positive test in the community) - this was often the most challenging category for acute trusts under covid pressure to collect the necessary data on. I don't think this is mentioned anywhere in the paper, though we had significant healthcare-associated community-onset outbreaks (eg. in community dialysis units, and among recently discharged patients re-admitted with Covid).

Response:

We agree that this is potentially helpful, though given that we don't make use of such a classification in the paper we decided not to include it in the paper. Note that, to our knowledge, this definition is only used in the UK, whereas the definitions we use and refer to are the standard ECDC definitions used across Europe (see ref 14: <https://www.ecdc.europa.eu/en/covid-19/surveillance/surveillance-definitions><https://www.ecdc.europa.eu/en/covid-19/surveillance/surveillance-definitions>).

Introduction page 5 lines 2-3: "understand its dynamical consequences." This could be clarified. Before reading the rest of the paper, it is not clear what 'dynamical consequences' means. Perhaps it could be rephrased to more plainly state the authors' aims for the introductory paragraphs.

Response: We agree that this was potentially confusing and have modified the text by deleting the first use of the phrase “dynamical consequences” on page 4 (since the last two sentences of the paragraph already explain what we mean, we don't think this takes away anything important), and rephrasing the last paragraph of the introduction as suggested.

I do not believe the authors state why they selected 10th June 2020 - 17th February 2021 for the study. This would be worth including, in introduction or methods.

Response:

We now give this information on lines 796-798 explaining that 10th June is the first date that information on classification as probable or definite healthcare associated based on time from admission to onset using ECDC criteria was consistently available for all trusts. Inevitably, the stop date is more arbitrary: we chose to take it as one month after vaccine rollout to the over 70s and the clinically extremely vulnerable began, as we judged that before this time patient vaccination was unlikely to have had a substantial impact on dynamics (vaccination status of admitted patients was not available to us and we therefore did not want to include periods where a large proportion of the patient population were likely to have vaccine-derived immunity).

Can the authors comment on why newer hospitals were associated with more hospital-acquired COVID-19? I would not have predicted this and find it very surprising. In my experience, older hospitals tend to have worse ventilation and more crowded wards, with fewer side rooms. Could this signal have been caused by confounders?

Response:

Anecdotally, we have heard suggestions that older hospitals may tend to be “more leaky” with higher ceilings than more modern buildings, but we have not been able to find any data on ventilation and building age so would rather not speculate on this in the paper. We do note, however, that while single room provision and heated volume/bed were both consistently associated with quite large reductions in transmission risk, this is not true for hospital age, where credible intervals for the IRR include 1 (SI, tables S3 and S6) and it is therefore not unlikely that this is a chance association. We have now modified the main text (page 10) so that we now report these credible intervals which were previously only given in the SI. It is certainly possible that none of the associations reported are directly causal, and we have been careful in the choice of language when discussing whether associations are causal or not (preferring to discuss the strength of evidence for causal associations particularly in light of results using the negative controls).

Can the authors expand on what ‘heated volume per bed’ is and how this relates to ventilation? I think demonstrating that a lack of side rooms and poor ventilation drives healthcare associated infections, ultimately causing patient harm and exacerbating the pandemic, is a very important outcome of this study. It would be good to highlight this more in the discussion.

Response:

This was a suggestion from Prof Cath Noakes (who we acknowledge), when we asked her about ventilation data which we were told was not available across trusts. As Prof Noakes pointed out “a higher volume per bed speaks to more air as well as better distancing”. It is simply the heated hospital building volume (from the NHS Estates Returns Information Collection) divided by the number of beds. We now explicitly highlight this result at the end of the first paragraph in the discussion (page 17) and refer to Prof Noakes’ modelling work on transmission of airborne infections in enclosed spaces (doi:10.1017/S0950268806005875) using the Wells–Riley equation which predicts that increased room volume should be associated with reduced transmission. The definition of heated volume (from Estates Returns Information Collection (ERIC) website <https://digital.nhs.uk/data-and-information/publications/statistical/estates-returns-information-collection>) is:

“The Heated Volume is the total gross internal floor area which is heated, multiplied by the height between the floor surface and the room ceiling, minus the floor area covered by internal walls and partitions (taken as 6%). Heated Volume should

exclude unheated spaces such as plant rooms, boiler houses, ceiling voids, pipe ducts, covered ways etc.”

The authors use a probability distribution for symptom onset by days since infection, with the most probable symptom onset day being 5-6 days post infection (Figure 1a). It would be interesting to add to the discussion how infection parameters such as this may differ between variants - in this study I suspect the distribution for Wuhan and Alpha variants will differ; I suspect Delta and Omicron would differ even more from Wuhan than Alpha. How might these changes affect the results - ie. how generalisable are the IPC recommendations across subsequent/ current/ potential future SARS-CoV-2 variants? The effect of Alpha was so strong (eg. figure 3a), what do the authors think of transmission with Delta and Omicron? Would the authors expect similar improvements to rates of hospital-acquired infections for other respiratory viruses such as influenza, if their recommendations were followed?

Response:

We agree that these are all interesting questions and a lot more could be added to the discussion if space permits, but at the current word count we would need to make substantial cuts to other parts of the manuscript to accommodate such an extended discussion. We note that the best analysis we are aware of suggests that generation times for the Alpha variant are similar to those for the Wuhan variant, suggesting that distribution to symptom onset may also be similar <https://elifesciences.org/articles/75791>

Re: the section using community-onset COVID-19 infections as a negative control, to check for confounding with the hospital-onset associations (page 10, lines 4-9). In my experience, the incidence of hospital-onset SARS-CoV-2 was strongly related to the incidence in the community, for multiple reasons including exhausting side room capacity limiting our ability to isolate patients, and presumably simply more opportunities for within-hospital transmission if there is more SARS-CoV-2 virus around, eg. via HCW infections picked up in the community being introduced to the hospital. Can the authors explain how controlling for community prevalence wouldn't eliminate legitimate signal, given the expected correlation between community prevalence and rates of hospital acquired infections? (In particular for the caveat given on heated volume per bed and HCW vaccination associations with healthcare-associated transmission).

Response:

There are two issues here. First, in all models where the dependent variable is either healthcare associated infections in patients or infections in HCWs we do indeed find (as suggested by the reviewer) that in all cases there was a positive association (“legitimate signal”) between these community-onset infections (in these cases occurring as independent variables, the d terms in Tables S3 to S14 in the SI) and infections in the hospitals, even after adjusting for other covariates. This is consistent with the influx into the hospital of patients with community-acquired COVID19 causing (directly or indirectly) more transmission in the hospital. There are

also models where we took community-acquired Covid-19 infection as the dependent variable. As the text above Table S15: Model P1.1.1 (page 27 of the SI) says “This is a negative control in the sense that many factors that might be supposed to causally affect the rate of hospital transmission in a trust (such as factors related to the hospital buildings) would not, in general, be expected to have a large impact on the admissions of community-acquired cases. For such covariates associations of similar magnitude and in the same direction for both the outcome of interest and for the negative control outcome can therefore provide evidence against there being a direct causal effect.”. This is by no means a perfect negative control (as pointed out above some community acquired infections may in fact have been acquired in hospital) and for this reason we don’t think the association of this negative control with heated volume per bed and HCW vaccination rule out important causal effects of these factors on healthcare-associated infection (and of course there are important reasons for thinking that they might play a causal role) , but they do indicate that we should be more circumspect in the interpretation of these results than we would have been if no association with the negative control had been found. We hope that this nuance comes across in the paper, but would be happy to rephrase if it does not.

I generally expect a scientific paper to include discussion of the study limitations in the Discussion, and how these may impact on the conclusions/ were addressed.

Response:

We have extended the consideration of limitations in the discussion and we now discuss what we consider to be the main limitations:

- i) lack of genomic data making it impossible to conclusively demonstrate transmission (line 366)
- ii) Lack of a national record keeping for when screening went beyond that nationally mandated (line 382)
- iii) Lack of centrally collected data on trust-specific IPC measures (line 383)
- iv) Lack of data concerning vaccination of the patient population from 8th December 2020 when vaccine was first used in the over 80s (line 389)
- v) Lack of PCR sensitivity estimates specific to the Alpha variant or conditioned on symptoms having occurred (line 388).

Limitations regarding ascribing causal interpretation to results are also considered in the results section (which seems more natural than considering this in the discussion) (page 11).

Other limitations have been addressed by additional sensitivity analysis and we have now added i) a model with time-varying intercept, ii) results with different assumptions about PCR sensitivity, iii) results with different criteria for selecting patients at risk of infection when determining length of stay distributions.

Some references that could be included in the relevant text:

- *Air filtration and airborne pathogen transmission (including SARS-CoV-2) in hospitals: Conway Morris, A., et al. 2021. Clinical Infectious Diseases, ciab933, <https://doi.org/10.1093/cid/ciab933>*
- *Vaccination reducing HCW infections for SARS-CoV-2: Ferris M, et al. 2021. Elife. 2021 Nov 16;10:e71131. doi: 10.7554/eLife.71131*

Response:

Thank you for these suggestions. We are familiar with both of these papers and have added a reference to the first.

Referee #4 (Remarks to the Author):

The COVID-19 pandemic has impacted healthcare facilities worldwide, causing large outbreaks with major morbidity and mortality. However, the actual number of hospital-acquired cases has rarely been quantified at a large scale, notably due to imperfect testing within healthcare settings, especially over the first epidemic waves. In this context, this paper proposes an assessment of the full burden of hospital-acquired SARS-CoV-2 in England between June 2020 and March 2021, identifies the main factors associated with nosocomial SARS-CoV-2 transmission to patients and healthcare workers, and studies the role played by this transmission on the efficiency of lockdown measures in the community.

The authors should be commended for their thorough exploration of the burden of hospital-acquired SARS-CoV-2 in England, combining large-scale data analysis, statistical modelling and dynamic mathematical modelling. This work clearly has important public health implications and should prove key in justifying further studies to help control SARS-CoV-2 transmission within healthcare settings.

There are however several points that I think should be addressed to benefit the paper.

Response:

Thank you for these encouraging comments and for the helpful suggestions below.

Regarding the burden quantification

Although detailed, the description of the method to quantify the number of hospital-acquired SARS-CoV-2 infections is at times unclear and hard to follow. This is, I believe, due to several reasons:

- The use, at times, of different terms for the same concept (e.g., “distribution of times from infection to onset of symptoms” vs. “incubation period distribution”);

- A lack of explicit equations in some parts of the section to help the reader understand the approach, rather than general sentences e.g. “In general, P_{ij} will depend on...”;

- Several repetitions, e.g. the mention that the definition of “definite” HAIs requires the first positive sample to be taken 15 days or more after hospital admission, which is made at least twice in the same section; or the definition of γ_{ijmd} which is provided twice, using slightly different terms, on the same page;

- Some confusing notations, e.g. on lines 6 and 8 of p. 23 π_{ij} is used to denote both the probability that a new nosocomial infection in trust i and week j is detected and the probability that a new nosocomial infection in trust i and week j is detected and meets the HAI definition; maybe the latter should read π'_{ij} ?

Response:

We have now made changes to the methods section so that we use consistent terminology (using “incubation period” throughout), and have improved notation (using π'_{ij} to represent the probability that a new nosocomial infection in trust i and week j is detected and meets the HAI definition as suggested). We have also removed the duplicate definition of “definite” HAI and use explicit equations (reserving explanatory text for situations where some readers might find the motivation for the equations opaque).

Overall, I would recommend reorganizing this method section to start from the most general equation (how the total number of HAIs z_{ij} is computed from the observed number of HAIs y_{ij} and the probability π_{ij} that an occurring infection in trust i and week j will be detected), and then to go on to explain how π_{ij} is computed from y_{ijmd} and P_{ijmd} (again providing equations straight away), and then only explain how these two components are computed. Adding subtitles to clearly delineate each component's calculation could also maybe help.

Response:

We have now re-organised this section along the lines suggested.

In addition, while efforts were clearly made to explore some uncertainties in this quantification through parameter distributions and sensitivity analyses, this is not the case for the assumed PCR test sensitivity over time since infection, which is informed by the paper from Hellewell et al. Indeed, this study is based on data on self-performed tests in healthcare workers, who are likely to have less severe forms of COVID-19 than patients, for whom the sensitivity curve is used here. How would using PCR sensitivity over time data from other papers (e.g., the study by Kucirka et al.), which are based on healthcare-worker-performed tests on patients, impact the results? I believe this would lead to lower probabilities of detection before day 3, but higher probabilities of detection after day 7, with potential consequences on the overall estimates.

Response:

Thank you for this suggestion. We have now done this (a sensitivity analysis using PCR sensitivity from Kucirka et al.) and results are reported in the SI (Supplementary results section 2.1). The estimated numbers of hospital-acquired infections are very similar to those using the estimates from Hellewell et al. We refer to this new sensitivity analysis on page 7 of the main paper: “Similar estimates are

obtained when using more granular length-of-stay data and in other sensitivity analyses”.

Regarding the statistical analysis

The authors underline that “both heated volume per bed and HCW vaccination coverage show similar negative associations with the control outcome as for reported healthcare-associated infection outcomes”. However, no possible explanation is proposed for this finding. Could this be related to hospital type for instance, with higher vaccination coverage and more admissions of COVID-19 patients in large university hospitals? It could be interesting to include a paragraph discussing possible interpretations in the discussion.

Response:

It is certainly possible that associations with the control outcome are in part driven by differences in hospitals, and we note that hospitals with larger heated volume per bed also tend to have more single rooms, and this pattern is consistent across NHS regions (though this might just reflect that beds in single rooms tend to take up more space). We have added as Extended data Figure 1 a pairs plot which shows this information and other correlations between hospital characteristics. This also shows a correlation between heated volume per bed and the cumulative number of patients with community acquired infection admitted to the hospital, though this correlation is not consistent across NHS regions.

We have added to the existing sentence considering such confounding (page 11) so that it now reads: “If associations between hospital characteristics (exposures) and this control outcome are similar to those for hospital-acquired infections, it would suggest that confounding is a plausible explanation for observed associations with hospital-acquired infections (for example due to differences in hospital characteristics not accounted for in the model).” We also now refer to a new Extended Data Fig. 9 which provides more explanation of how the negative control outcome can help distinguish between different causal relationships.

Regarding the mathematical modelling

A general description of the model is lacking from the methods, which only includes a description of the scenarios explored, followed straight away by the model equations. Why include these equations in the Methods but put the model diagram and lists of model variables and parameters in the Supplement? I’m afraid that these equations are impossible to understand without these lists. I would suggest putting the model diagram in the Methods, along with a paragraph briefly describing the main features of the model (main populations involved and main disease stages

considered, who is assumed to be in contact with whom etc.), but moving the equations to the Supplement.

As a minor remark, the equation on SH is written twice on p.31.

Response:

We have made these changes as suggested and removed the duplicated equation (note that equations now appear in the SI).

Reviewer Reports on the First Revision:

Referees' comments:

Referee #4 (Remarks to the Author):

Thank you for the additional work performed, including the new sensitivity analyses. I believe that the manuscript is improved and that my concerns have been addressed in a satisfactory manner.

Referee #2 (Remarks to the Author):

The authors have addressed most of my questions. I have a couple of minor questions/suggestions:

1. Some important quantities are not easy to understand, or their definitions sound confusing, and a bit more clarification is recommended. For example, in the section “effect of testing policy”, it states “ P_{ijmd} is the probability that a patient admitted to trust i in week j ...” which sounds like the patient was admitted to trust i exactly during week j . In the definition of γ_{ijmd} , “given a new hospital-acquired infection in trust i in week j occurs”, which sounds like the infection occurred in week j . I think neither is necessarily the case. The whole purpose of defining these quantities is to estimate z_{ij} , the true number of active infections (which can test positive if tested) in trust i during week j , based on the observed number of hospital-acquired infections, y_{ij} , which is actually the number of PCR-positive patients who have been admitted for at least 15 days. That is, actual infection of this patient did not necessarily occur during week j , and admission definitely should not have occurred during week j . It’s a little tricky to make all these clear, but I think clarification is necessary.
2. It is still not clear why it is justifiable to approximate γ_{ijmd} by ψ_{imd} . The definitions of the two quantities are distinct. γ_{ijmd} is “for a given hospital-acquired infection in trust i during week j , the probability that this patient has a length of stay m and infection was on day of stay d ”, whereas ψ_{imd} is “on a given day, the probability a patient with a length of stay m is on day of stay d ”. This approximation may be OK if the infection hazard is low over time but may not work well otherwise.
3. Line 581, “a randomly chosen day m a randomly chosen patient...”, m should be removed.
4. Line 126, “with 12% of all such infections”, by “such infections” do you refer to “all hospital-acquired infections” or “all detected hospital-acquired infections”?
5. Line 235-236, “the number of patients admitted with community-acquired SARS-CoV-2 infection”, is it based on the ECDC definition? Please make it clear.
6. Any implication of this study in the Omicron age?

Response to reviewers' comments

We would like to thank all the reviewers for their careful consideration of the revised manuscript. Below we outline how we have addressed the remaining minor questions/suggestions.

1. Some important quantities are not easy to understand, or their definitions sound confusing, and a bit more clarification is recommended. For example, in the section "effect of testing policy", it states " P_{ijmd} is the probability that a patient admitted to trust i in week j ..." which sounds like the patient was admitted to trust i exactly during week j . in the definition of γ_{ijmd} , "given a new hospital-acquired infection in trust i in week j occurs", which sounds like the infection occurred in week j . I think neither is necessarily the case. The whole purpose of defining these quantities is to estimate z_{ij} , the true number of active infections (which can test positive if tested) in trust i during week j , based on the observed number of hospital-acquired infections, y_{ij} , which is actually the number of PCR-positive patients who have been admitted for at least 15 days. That is, actual infection of this patient did not necessarily occurred during week j , and admission definitely should not have occurred during week j . It's a little tricky to make all these clear, but I think clarification is necessary.

We agree – the terminology is confusing. In an attempt to describe the methodology in the most general terms, allowing for testing policies and length of stay distributions that vary both with time and trust, we have created unnecessary complexity. As the reviewer rightly points out the infections z_{ij} will not necessarily have occurred in week j , but could have occurred one, two or more weeks previously, and explaining this is indeed quite tricky with plenty of potential for confusion and requiring lots of extra verbiage. Since the primary analysis reported in the paper is not concerned with the time varying nature of hospital-acquired infections and uses a length of stay distribution aggregated over the whole period, we think, on reflection, that the simplest approach to increase clarity is to drop the j subscript. Having explained the approach in this simple case (where the probability that nosocomial infections are detected and correctly identified does not vary with time), it is a simple matter (without requiring additional cumbersome notation) to see how it generalises to the situation where we use different length of stay distributions for different time periods, which we explore in sensitivity analyses, or to situations where the testing policy changes over time. Note that we need to keep the j subscript in the methods section for "Quantifying drivers of nosocomial transmission" as this part of the analysis explicitly accounts for time-dependence.

We have thus made the following changes:

- i) In the methods we have dropped the j subscripts in the section describing the methods for quantifying the number of hospital-acquired infections (lines 539-603).
- ii) In the caption to figure 1 we now refer to the green and blue lines as "the estimated total number of hospital-acquired infections across adult acute

NHS trusts in England linked to observed weekly number of detected post day 14 onset infections,”

2. It is still not clear why it is justifiable to approximate γ_{ijmd} by ψ_{imd} . The definitions of the two quantities are distinct. γ_{ijmd} is “for a given hospital-acquired infection in trust i during week j, the probability that this patient has a length of stay m and infection was on day of stay d”, whereas ψ_{imd} is “on a given day, the probability a patient with a length of stay m is on day of stay d”. This approximation may be OK if the infection hazard is low over time but may not work well otherwise.

Yes, they are distinct quantities but, as the reviewer indicates, the approximation (which amounts to an assumption that, given a patient becomes infected, he or she is equally likely to be infected on each day of his or her stay) will be reasonable provided the infection hazard is low. As a numerical example, consider a patient with a length of stay of 20 days subject to a constant hazard of infection, and with an overall probability of becoming infected of 1%. Then, given infected, the conditional probabilities of being infected on day 1, day 2,....day 20 are:

0.0502 0.0502 0.0502 0.0502 0.0501 0.0501 0.0501 0.0501 0.0500 0.0500 0.0500
0.0500 0.0499 0.0499 0.0499 0.0499 0.0498 0.0498 0.0498 0.0498.

For a hazard corresponding to an overall probability of becoming infected of 2% the corresponding conditional probabilities are:

0.0505 0.0504 0.0504 0.0503 0.0503 0.0502 0.0502 0.0501 0.0501 0.0500 0.0500
0.0499 0.0499 0.0498 0.0498 0.0497 0.0497 0.0496 0.0496 0.0495

For an infection hazard that would result in 2% of all patients becoming infected (assuming all are susceptible on admission and accounting for the empirical length of stay distribution) the conditional probabilities of being infected on day 1, day 2,....day 20 (for a patient with length of stay of 20 days who does become infected) are still approximately constant and declining only slowly (

0.0517 0.0515 0.0513 0.0512 0.0510 0.0508 0.0506 0.0504 0.0503 0.0501 0.0499
0.0497 0.0495 0.0494 0.0492 0.0490) indicating that the approximation is reasonable for the range of infection rates seen in hospitals in England.

Code supporting these calculations can be found here:

https://github.com/BenSCooper/nosocomial_COVID_England/blob/code-for-resubmission/calc%20prob%20infected%20by%20day%20of%20stay.R

We have also added the text (line 606): “This will represent a reasonable approximation provided that the infection hazard is small and approximately constant over a patient’s hospital stay.”

- 3 Line 581, “a randomly chosen day m a randomly chosen patient...”, m should be removed.

Thank you. This has been fixed.

- 4 Line 126, “with 12% of all such infections”, by “such infections” do you refer to “all hospital acquired infections” or “all detected hospital-acquired infections”?

The former. The text has now been changed to “with 12% (10%, 14%) of all hospital-acquired infections meeting criteria for definite healthcare-associated infection” to remove this ambiguity.

- 5 Line 235-236, “the number of patients admitted with community-acquired SARS-CoV-2 infection”, is it based on the ECDC definition? Please make it clear.

Yes. This has now been clarified in the text.

- 6 Any implication of this study in the Omicron age?

We have addressed this by adding the following sentence to the discussion:

“While our analysis focuses on nosocomial transmission early in the pandemic and prior to widespread vaccine coverage, the subsequent emergence of the highly contagious Omicron variants of SARS-CoV-2 presents additional infection control challenges, with high rates of hospital-onset infection reported despite high vaccine coverage, universal masking, admission testing, and symptom-based screening; anecdotal reports suggest that heightened control measures may be needed to suppress nosocomial spread [<https://doi.org/10.1093/cid/ciac113>].”

Reviewer Reports on the Second Revision:

Referees' comments:

Referee #2 (Remarks to the Author):

The authors have addressed all my questions.